# Stability Analysis of the Tailings Dam for the Purpose of Closing, Greening, and Ensuring Its Safety—Study Case

Mihaela Toderaș [1,*], Vlad Alexandru Florea [2] and Răzvan Bogdan Itu [2]

1   Mining Engineering, Surveying and Civil Engineering Department, Faculty of Mines, University of Petrosani, 332006 Petrosani, Romania
2   Department of Mechanical, Industrial and Transportation Engineering, University of Petrosani, 332006 Petrosani, Romania; vladflorea@upet.ro (V.A.F.); razvanitu@upet.ro (R.B.I.)
*   Correspondence: toderasmihaela@yahoo.com; Tel.: +40-741-501-143

**Abstract:** Tailings dams are special constructions that are part of a complex of works related to the installations for preparing mining masses. These constructions play a role in the mechanical treatment of wastewater and the safe storage of sterile material resulting from ore processing. The closing and greening of tailings dams is achieved taking into account the general stability of the pond and its related constructions, as well as the integration into the surrounding environment of the surfaces occupied by mining waste deposits (tailings dams). This study presents the results of the hydrogeotechnical and stability study carried out on the Gura Roşiei tailings dam location. This analysis aims to evaluate the stability degree of the three compartments that comprise the tailings deposit in order to carry out the closing and greening works of the tailings dam, and to conclude whether the idea of raising them by 1.5–2 m with tailings is feasible. This study was based on field observations, geotechnical drilling, and physical–chemical analyses of the collected samples. Due to the shallow depth, the drilling could not highlight a hydrostatic level except for the areas with excess humidity, areas represented by certain lenticular intercalations of sandy dust that yield water very slowly. These areas are not continuous and cannot define a reference hydrostatic level. All situations were analyzed by four different methods that satisfy the static equilibrium of forces or moments (Bishop and Janbu) or simultaneously of forces and moments (Spencer and Morgenstern–Price). From the point of view of the stability calculations performed in the hypothesis in which the three ponds become active for the storage of tailings, assuming a corresponding piezometric level, the resulting safety factors are relatively close to the standard values (Fs $\geq$ 1.4) for the static analysis, and in seismic conditions, they are at the limit of equilibrium. The NE slope of pond No. 2 shows values below the standard safety limit for this type of work. Moreover, tailings dam No. 3 presented from the calculations as being totally inadequate for the elevation. From the obtained results, it was found that the location formed by the three compartments that comprise the Gura Roşiei tailings dam presents major disadvantages for a future storage of flotation tailings, being at the same time an imminent danger to the environment. Due to its reduced capacity to release water from its pores, the settled material is still in a saturated state, and it is assumed that the foundation land, comprised of the terrace deposits of the Abrudel River, is clogged at the interface with the settled material and unable to naturally drain the excess moisture from the dam's body.

**Keywords:** tailings dam; stability; mining waste deposits; safety factor; greening; environment

## 1. Introduction

Mining activity generates an impact that affects environmental factors by producing a significant volume of waste [1–4]. Depending on a series of variables, the impact can be permanent or temporary, reversible or irreversible, and negative or positive. On the other hand, the mining industry has the ability to utilize the techniques and technologies necessary for the ecological rehabilitation of degraded lands and for the elimination of the

impact, regarding the practice of responsible mining, focused on the three basic elements of sustainable development: economic development, environmental protection, and social protection. By adding technological progress to these basic elements, mining can become a sustainable activity in the medium and long term. As a whole, mining activities do not fit indefinitely into the context of sustainable development. However, the sequence of mining operation—exploration, opening, exploitation, closure, and ecological rehabilitation—can be directed so that the environment, the economy, and the local community reach higher quality standards than before the development of mining [3–8].

Ensuring the stability of mining waste deposits (dumps and tailings dams) represents one of the most important problems in mining activity, both from a technological and environmental protection point of view, as a possible slippage can have particularly serious consequences. The design problems of tailings dumps and dams differ depending on the characteristics of the material to be stored, the characteristics of the location and the method of deposition, and the volume of waste material to be stored, particularly its height and extent. For the control of tailings deposits under construction or conservation, from the design phase on, it is necessary to provide a series of measures related to the periodic control of the stability of the tailings deposit and the surrounding areas and to carry out geotechnical studies on the rocks in the foundation and on the rocks and substances in the tailings deposit [7–10].

Due to the fact that many tailings deposits (dumps and dams) have been damaged, leading to catastrophes of large proportions that have affected extensive territories, it is absolutely necessary to carry out a large number of in situ and laboratory studies, which serve to monitor their technical state [1,3,4,11–18]. In order to protect the natural and anthropogenic environment in the area of influence of the dumps, field studies are necessary, from which the following elements can result: the possibility of occupying the smallest and least productive land surfaces; the influence of residues on the environment; the possibility of entrainment of particles by the prevailing winds; landfill stabilization measures to eliminate negative effects; the delimitation of the areas of influence and the establishment of the minimum distances between the tailings deposits and inhabited areas; and, after abandonment, there should be the possibility of inclusion into the general landscape, so that there are no dissonances between the natural areas and those altered by the tailings deposits.

The tailings dams represent a natural configuration or technical arrangement for the storage of fine-grained waste, mainly processing tailings, together with variable amounts of free water, resulting from the processing (preparation) of mineral resources and from the clarification and recirculation of processing water.

As a result of mining activities, over time in Romania, 64 tailings dams were built, which occupy an area of almost 1350 ha and store over 350 million m$^3$ of tailings. The negative effects generated on the environment by the construction and operation of tailings ponds can be summarized as follows: unpleasant visual impact; the destruction and occupation of large areas of land for a very long period of time [1,2,12–14]; pollution of surface or underground waters with dissolved chemical elements or suspensions of solid particles entrained from dikes by rainwater or seepage; air pollution with dust powder and gases resulting from the minerals contained in the dams or produced by their oxidation and combustion; material destruction and loss of human lives due to the loss of stability, etc. [16,18–20].

The tailings dams for tailings resulting from the activity of ore treatment in the Apuseni Mountains are usually located in the valleys or in the river meadows in the area. They have heights of 20–40 m and even more, and each of them occupies an area of tens of hectares. In the zonal morphology, these engineering constructions appear as positive forms of relief that contrast with the flatness of the meadow or valley relief. The annual volume of flotation tailings that was stored at the country level in the tailings dams exceeded 5 million tons, and the volume of water discharged into the emissary amounted to approximately 50 million m$^3$. The near-universal use of tailings dams as the primary method of storing tailings and treating polluted waters in the mining industry undoubtedly stems from the

fact that they can fulfill several functions and represent a system of storage and treatment of tailings and waters that is quite cheap.

Mines that use well-constructed and permanently controlled tailings dams easily obtain effluent that very well respects the limits imposed by the legislation in force. In general, in the immediate vicinity of the ponds located in narrow valleys, siltation phenomena appeared as a result of the local water level rise, and landslides occurred on the slopes caused by the excess moisture created by blocking the water horizons. At the dams built on flat lands, in the river meadows, overflowing phenomena have been installed, and near them, there are accumulations of water that have produced silting and salinization phenomena. These are due in large part to exfiltration through dikes [20–23]. In his studies, Burd [24] analyzes changes in tailings thickness and copper levels before, during, and after mining and highlights three distinct impact zones below the dumping depth. The issue of the irrational use of natural resources in Ukraine that affects public health, population working ability, and macroeconomic performance was study by Koval et al. [25]. The key to the study is the formation of a holistic view of the relationship between pollution and the state of the environment and harm to public health based on the analysis of rational nature management and environmental pollution and their negative impact on environmental health [25]. Due to the fact that no basin is completely exempt from exfiltration, it is important to control not only the quantity but also the quality of exfiltration [25,26]. In this regard, it should be pointed out that, recently, filter and drainage devices built into the dikes have been used, which facilitate the capture of exfiltrated water for its purification before it reaches local and regional receivers.

Three fundamental aspects must be taken into account when discussing the issue of tailings dam safety. The first one refers to physical stability, i.e., the breaking of the dike due to some mechanism such as circular sliding, sinking, regressive erosion when water spills over the dike, etc. The second aspect concerns the chemical stability, which refers to the increase in water acidity (the presence of sulfides in the material deposited in the pond, especially pyrite), which draws heavy metals into solution. The third aspect refers to the fact that the loss of toxic material through exfiltration must be taken into account.

The design, construction, and scientific research of tailings dams have made significant progress in recent decades. In [27], the possibilities and limitations of such models are discussed with the purpose to give geotechnical engineers (rather than researchers) guidelines to properly select soil models and their corresponding parameters to be used in the finite element method for engineering applications. However, maintaining the ponds in a state that avoids pollution requires continuous and multidisciplinary research [28–30], with the participation of engineers, geologists, agronomists, foresters, chemists, etc. The frequency of accidents due to the instability of the waste dump slope structures has also increased, resulting in significant fatalities, apart from the economic, social, and environmental impacts of these disasters. Numerous scientific studies have been conducted to reduce the occurrence of such incidents [31].

The closure and greening of tailings dams is carried out while considering the general stability of the pond and its related constructions, as well as the integration into the environment of the surfaces occupied by mining waste deposits (tailings ponds). The tailings ponds and tailings dumps in Alba County present an immediate danger in terms of the occurrence of ecological accidents. In this situation, the tailings dam from Gura Roşiei is under the administration of S.C. Branch RosiaMin, S.A. Rosia Montana. The Gura Roşiei tailings dam, on which there was a hydraulic screen, presents a high degree of risk regarding the possibility of accidents, with a particular impact on human lives and environmental factors.

The purpose of this work is to conduct a stability study of the Gura Roşiei tailings dam in order to carry out the works for its closure and greening, given the fact that the mining activity was stopped in 2006. The planned works have the purpose of improving the management and safety of the tailings deposit and the impact produced by its existence on the environmental factors in the area. The implementation of the planned works will lead

to a reduction of the risk coefficient, the disappearance of some sources of environmental pollution, and a reduction in the intensity of others.

The need to analyze tailings dams 1, 2, and 3 was thought to be related to the idea of having a storage space for the tailings of the preparation in the event of the resumption of gold–silver ore processing activity in the area. For the arrangement and greening of these ponds in order to restore them to the natural circuit and fit them into the landscape of the area, it is absolutely necessary to carry out a stability study. The results of this study formed the basis for the choice of solutions, methods, and materials used in the greening and safety projects at the Gura Roşiei dam.

## 2. General Data Regarding the Tailings Dam

The tailings dams are special hydrotechnical retention constructions of the permeable type. These represent the complex of works related to the mining mass preparation installations, with the aim on the one hand of mechanical purification of wastewater and on the other hand of safe storage, in general, for an unlimited or indefinite time, of the tailings resulting from the preparation of ores. The term "tailings dam" is used only in cases where the tailings are discharged from the preparation facilities in the form of a tailings slurry, in which the solid part is fine-grained, the grains of material being generally well below 1 mm in size.

The tailings dams No. 1, 2, and 3 of Gura Roşiei are located on the left bank of the Abrudel stream, about 1 km upstream from the Gura Roşiei processing plant. The complex of tailings dams from Gura Roşiei, which served the gold–silver ore preparation plant from Rosia Montana, is made up of three compartments that functioned successively and in alternation with the Valea Selistei tailings dam. They came into operation as follows: in 1961, No. 1 dam; in 1966, No. 2 dam; and in 1974, dam No. 3. The tailings dam No. 2 replaced No. 1, and the No. 3 dam replaced pond No. 2; the tailings dam No. 3 was exploited until 1988, and during the last two years of operation, it worked alternately with the Valea Selistei tailings dam (put into operation in 1986). Dam No. 1 has a height of 15 m, dam No. 2 of 45 m, and dam No. 3 of 42 m, together having stored a quantity of almost 10 million tons of preparation waste. Gura Roşiei dams No. 1, 2, and 3 have served the mining company from Rosia Montana for over 35 years, during which time they stored the tailings resulting from the gold–silver ore preparation plant, thus becoming, due to a certain degree of recovery, an attractive reserve for possible subsequent reprocessing. In this paper, the stability study does not aim at this aspect. We specify the fact that the waste from the ponds is already ground and can be found in the immediate vicinity of the preparation plant. In the case of reprocessing, this location would result in reduced costs in terms of extraction, processing, and transport.

The tailings dams overlap morphologically over the terrace of the Abrudel stream and partially over the eastern slope of the Dăroaia hill. From a geological point of view, in the region, there are cretaceous formations that form the foundation of the region: they are made up of alternations of clays, marls, sandstones, and microconglomerates; quaternary deposits that form the covering blanket; and alluvial and diluvial deposits. The region is "cut" from south to north by the Abrudel stream and has as its tributaries the Seliste Valley, the Dăroaia Valley, and the Roşiei Valley; it in turn flows into the Aries River near the town of Câmpeni. From a seismic point of view, the region falls within the zone of six degrees.

### 2.1. Constructive Elements

The Gura Roşiei dam is a coastal pond with an inward elevation (Figure 1). At the level of the contour dam, it has an area of approximately 9 ha, and the maximum height of the tailings deposit is roughly 45 m.

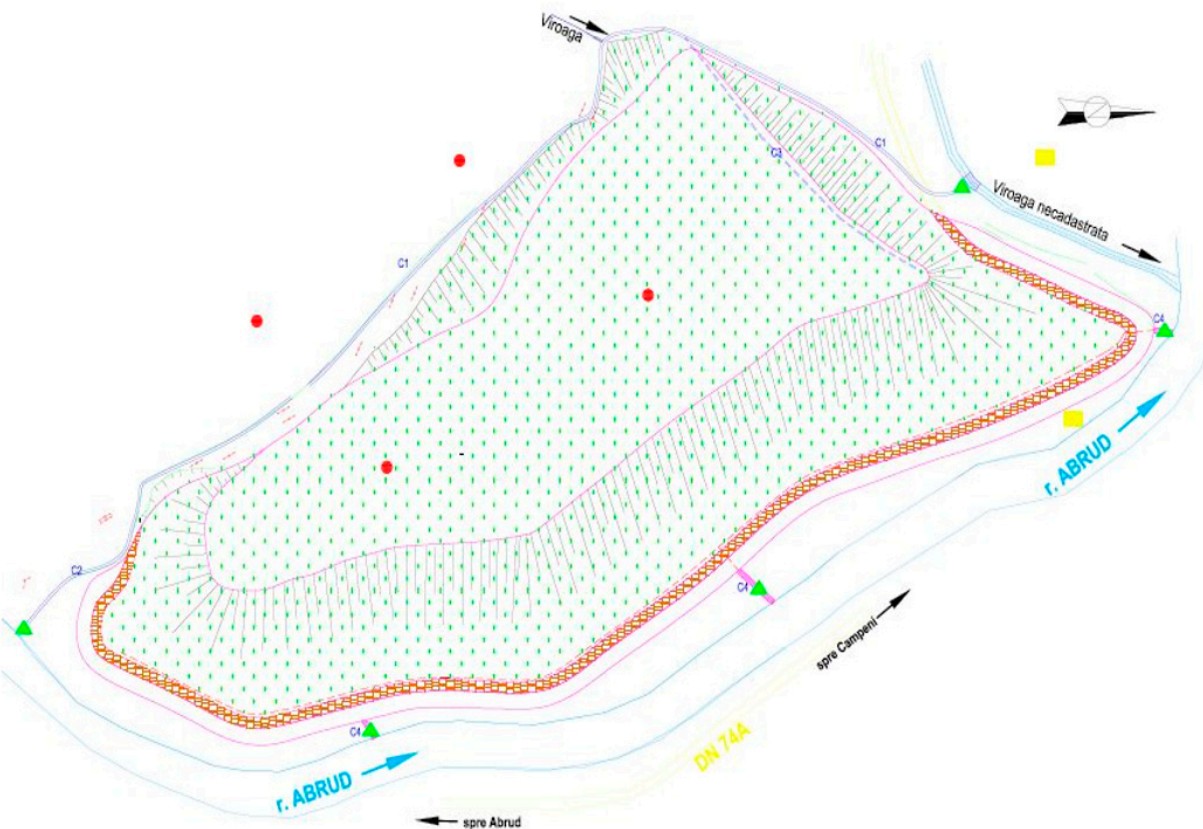

**Figure 1.** Situation plan of the tailings dam.

The constructive structure of the pond includes the following components:

- contour dike (priming): It was built on two sides of the tailings dam, namely on the NW side and on the side parallel to the Abrudel stream. The other sides of the dam are natural slope areas. The dike was made of stone and river ballast. Made in this way, it also fulfills the role of a "reverse filter", allowing the water to be filtered from the body of the pond. The constructive elements of the contour dam are the width of the crest (2 m); maximum height (5 m); and declivity of the slopes (approx. 1:1).
- the system of transporting and depositing the sterile material: It has the role of transporting and distributing the sterile material in all areas in which it is needed, for the uniform loading of the available storage volume, to ensure the material necessary to build the elevation dams, and to ensure the best conditions for water clearing.
- waste warehouse itself: This warehouse was formed behind the priming dam and has three compartments (1, 2, and 3). The first compartment has not been used for a long time and has remained at a much lower share than compartments 2 and 3 (Figures 1 and 2). Compartments 2 and 3 are in preservation and were used only in exceptional situations, when damage would occur in the Selistei Valley. The maximum height of the deposit in these compartments is about 45 m. It was made by the "elevation inward" method, and the elevation dams were built from deposited and sediment waste in the previous stage.

The chemical characteristics of the deposited waste in the Gura Roşiei dam are presented in Table 1 [32–36].

The slopes made from sterile material were strongly affected by erosion phenomena caused by the waters of streams (gutters, ogres, and even ravines) (Figure 3). Access to the pond was very difficult, even under favorable weather conditions. The access road had a very large slope, was not practicable in adverse weather, and constituted a way to concentrate superficial leaks, with negative consequences for the evolution of the erosions caused by water.

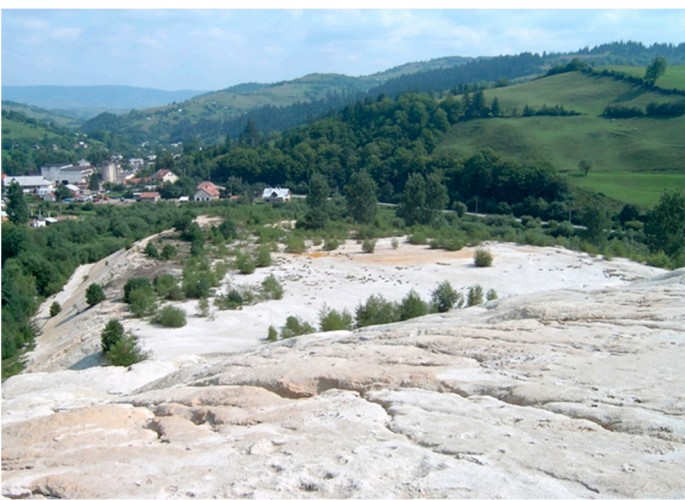

**Figure 2.** Gura Roşiei tailings dam, compartment no. 1.

**Table 1.** Chemical characteristics of the deposited waste in the tailings dam.

| Chemical Element | Value (%) | Chemical Element | Value (%) |
|---|---|---|---|
| $SiO_2$ | 65.31 | Pb | 0.06 |
| $Al_2O_3$ | 12.33 | Cu | 0.03 |
| Fe | 2.15 | Zn | 0.06 |
| CaO | 2.08 | V | 0.0003 |
| MgO | 1.49 | Cr | 0.0003 |
| S | 0.89 | Ni | 0.0003 |
| Mn | 0.10 | W | 0.003 |

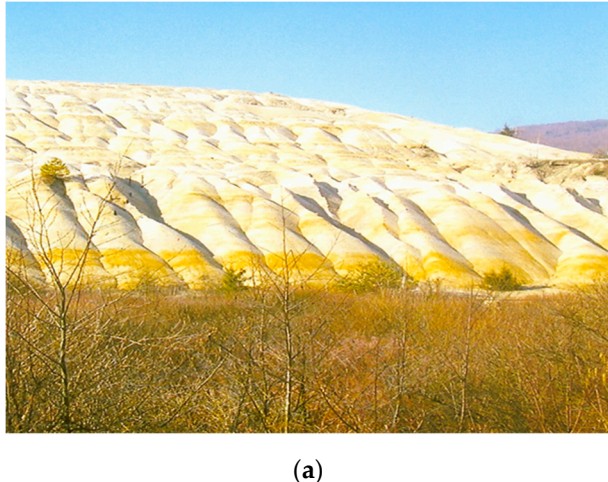

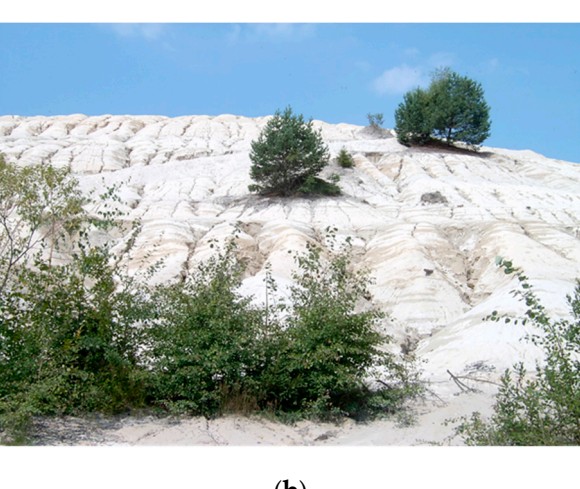

(**a**)  (**b**)

**Figure 3.** Erosion phenomena in the case of the Gura Roşiei dam: (**a**) along the slope; (**b**) in the area of the access road.

The upper platform of the pond is the former beach, with a slight inclination of approximately 0.5–1% toward the collection points of the cleared water. There were two such visible points in the form of two small lakes (Figure 4). The water in these lakes was relatively unpolluted. Around the platform of the pond, there was a raised dam with a 0.5 m guard compared to the beach and about 1.0 m compared to the cleared water level. At the foot of the slope, on the side parallel to the Abrudel stream, a quasi-dam was built, made of pond material, with the pond and transverse dams that form small compartments to retain the waste driven by the waters of the pond. Periodically, these compartments

are mechanically decolmatized, with the material resulting from these operations being delivered to ceramic enterprises (about 200 t/year) or transported on the pond platform.

- the drainage system: There is no information on the existence and functioning of this system.
- water evacuation system: For this pond, there was a unique system for the evacuation of water from rainfall and from the clearing slime pulp. This system is based on a reverse probe (Figure 5), which evacuates the water through an underground metal pipe (Ø = 400 mm) in the Abrudel stream (Figure 6). The pond was provided with quasi-parallel compartments to the Abrudel stream for the interception of tailings released by the seepage waters from the slopes of the pond, thus avoiding discharge into the emissary. Based on the observations made in the field, the following aspects were noted:

1. There were only two systems of reverse wells: one was in operation and the other was clogged with branches, stones, etc.
2. There was no access to the only working probe system, and the probe was not protected against clogging with floats.
3. The discharge pipe of the water captured by the reverse probe is metallic, underground, and has been in an acidic environment for a very long time.

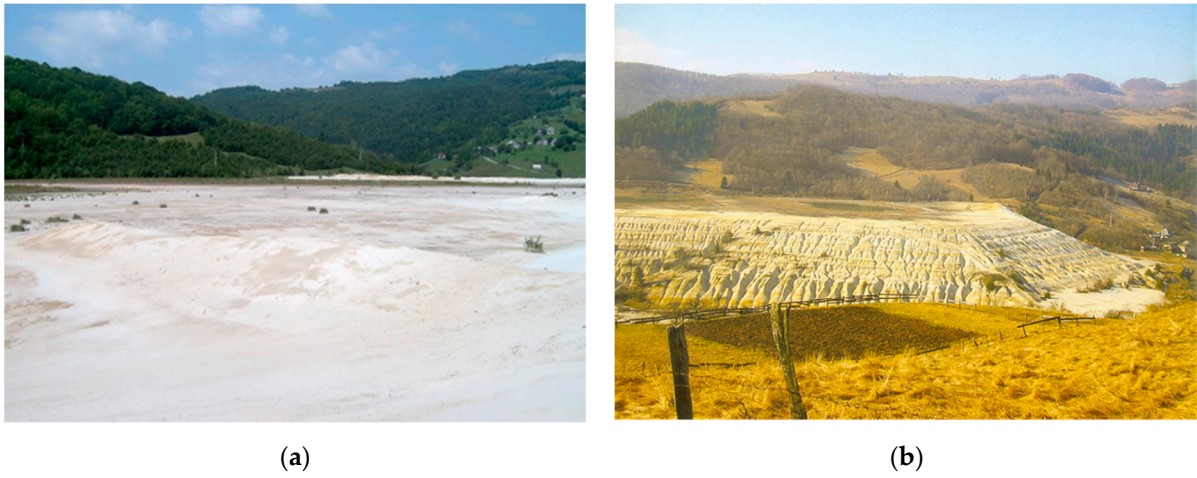

(**a**)    (**b**)

**Figure 4.** Lakes for the collection of clarified waters: (**a**) on the platform of the dam; (**b**) view from the eastern side of the dam.

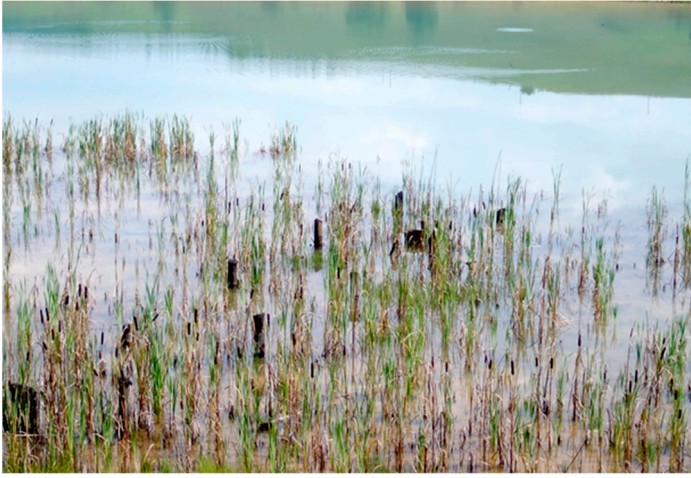

**Figure 5.** Reverse probe discharge of rainwater and clear waters.

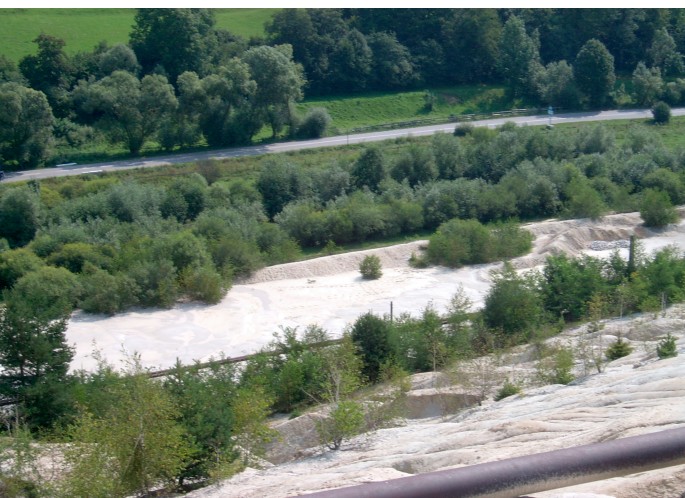

**Figure 6.** Gura Roşiei dam, waste retention compartments carried by rainwater.

They confirm the vulnerability of the water evacuation system from the Gura Roşiei tailings dam and attest to the fact that the pond is not safe in operation in relation to water evacuation.

- the system for monitoring the construction behavior: the Gura Roşiei tailings dam has class and category of importance II, respectively, B; there is no information about the system for monitoring the behavior of this tailings dam.

The materials deposited as a result of the activities carried out at the Gura Roşiei settling pond are sterile due to flotation and water. The Gura Roşiei tailings dam, currently inactive, represents a source of pollution by wind entrainment of suspended particles in dry periods, with the mass flow being estimated at approximately 6–14 kg/day/ha depending on the wind speed and the humidity state of the free surface. There are no NOx measurements in the area. Although the appearance of spontaneous vegetation is observed, active erosion of the slopes is still observed through the action of the seepage waters.

*2.2. Emission Sources*

The sources of emissions generated by the activities carried out at the Gura Roşiei tailings dam are as follows:

- water discharges: direct discharges of polluted water into natural surface receptors; exfiltration of polluted water from the pond into the groundwater in the area; direct discharge of stored waste into surface waters (occurred during current operation but can also occur in the event of an accident or outcome);
- emissions to the atmosphere: suspended dust blown by the wind from the material deposited in the tailings dam;
- direct discharge of tailings from the tailings dam onto the land surface located downstream of the settling pond (in case of accidents or damages).

The probability of an accident or breakdown is quantified by the stability coefficients established following the technical expertise carried out by the specialist designer regarding the safety of the tailings dam during operation. Correlating the results of these calculations with the findings in the field, we believe that this pond is in imminent danger. In relation to the possibility of damage or an accident at the settling ponds, it should be mentioned that much more serious than the impact on the environmental components is the impact on human settlements.

## 3. The Hydrogeotechnical and Stability Study of the Tailings Dam

To perform the stability study, 10 boreholes were made in the bodies of the ponds, from which disturbed and undisturbed samples were taken at different depths. These

samples were later analyzed in a geotechnical laboratory, obtaining the physical–mechanical characteristics necessary for stability and infiltration calculations. In addition to calculating the safety factors (stability coefficients) and the flow capable of exfiltration in the two hypotheses (in the current stage and in the hypothesis of an elevation of approximately 2 m), the determination of the bearing capacity of the land represented by the three settling ponds was also pursued and analyzed. The stability studies that were carried out in previous years were also analyzed in order to know the problems that these ponds had while they were in operation.

### 3.1. Methods Used in the Evaluation of the Safety Factor

Sakurai and Shimizu (1987) [37] proposed the use of fuzzy theory for the qualitative assessment of slope stability based on the uncertain factor of safety (Fs), defined as a trapezoidal membership function. Fuzzy logic tools allow the rigorous treatment of qualitative evaluation problems, having at their disposal a series of data with a high degree of approximation. The usefulness of the fuzzy method in the context of slope stability analyses lies in its ability to provide qualitative indications on the degree of slope stability with the help of a rigorous procedure. The method does not replace the quantitative, deterministic, and probabilistic methods, but it can represent a tool for primary estimation of the probability of sliding, thus being able to provide the designer, before carrying out the geotechnical research and its interpretation, with an image regarding the possible consolidation works [38–44]. In engineering practice, there are many methods for determining the safety factor for natural slopes, dams, etc.: Fellenius, Taylor, Bishop, Morgenstern–Price, Spencer, Janbu, the finite element method, statistical methods, and many others [45–50]. Most of them assume that the sliding surface is a circular surface, and the calculation algorithms are applied based on this assumption [51–53]. Johari et al. (2015) proposed two probabilistic models to assess the effect of seismic loading. They develop a probabilistic model of seismic slope stability based on Bishop's method using the JDRV method and make a comparison of the probability density functions of slope safety factor with the Monte Carlo simulation [54]. The same Monte Carlo simulation was used by the author [55] to compare the results obtained applying the jointly distributed random variables method for stochastic analysis and reliability assessment of seismic stability of infinite slopes without seepage. Considering the method of slices and by selecting stochastic soil parameters, the most critical failure surface is determined by the particle swarm optimization algorithm. The results are verified by Monte Carlo simulation (MCS); however, this method is quite time-consuming for this purpose [56]. Depending on the addressed hypothesis, these methods assume the solution of a system of equations from static mechanics that will satisfy the balance of moments and/or the equilibrium of forces of each vertical strip that represents the discretization element of a potential circular or non-circular sliding surface. Table 2 presents these methods together with the satisfied assumptions within the static equilibrium.

**Table 2.** Methods and hypotheses of calculus.

| Method | Equilibrium of Moments | Equilibrium of Forces |
|---|---|---|
| Fellenius | YES | NO |
| Bishop simplified | YES | NO |
| Janbu simplified | NO | YES |
| Janbu corrected | YES (on the strip) | YES |
| Morgenstern–Price (GLE) | YES | YES |
| Spencer | YES | YES |
| US Army Corps of Engineers-1 | NO | YES |
| US Army Corps of Engineers-2 | NO | YES |
| Lowe–Karafiath | NO | YES |

The conditions imposed by the static equilibrium will be applied for an individual strip as well as for the entire mass above the sliding surface. The system of forces acting on the four sides of a characteristic strip is shown in Figures 7 and 8.

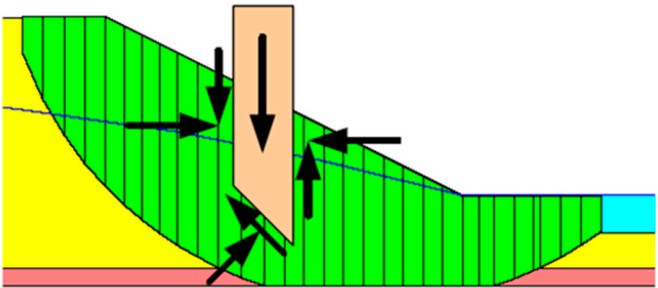

**Figure 7.** Dividing the sliding surface into vertical slices.

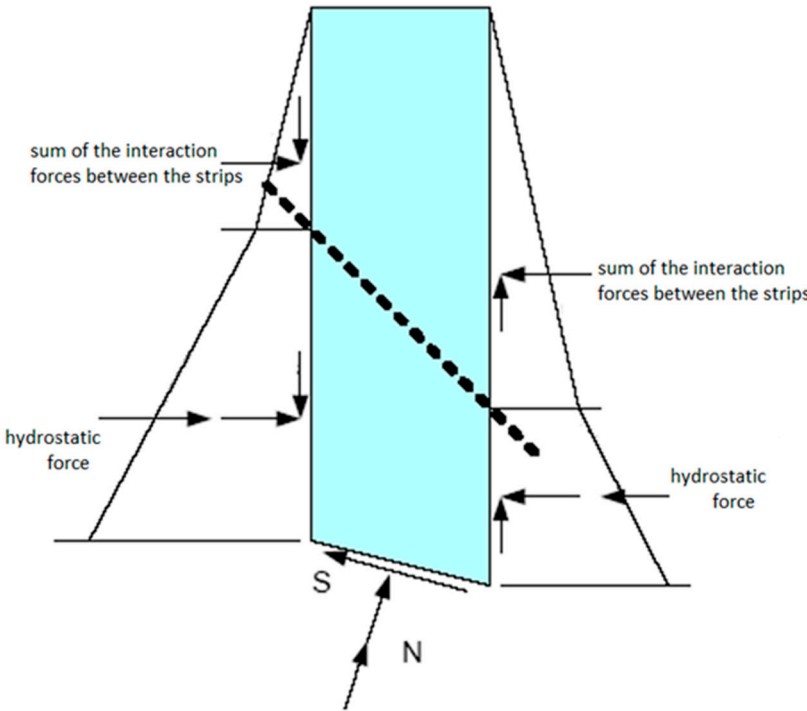

**Figure 8.** The system of forces acting at the level of a characteristic slice.

The conditions of static equilibrium involve solving the following situations:

1.  The static equilibrium of each strip is ensured by the equilibrium of the projections in two orthogonal directions and the equilibrium of the moments of the forces with respect to any point.
2.  Total vertical equilibrium is ensured when the vertical component of the resistance forces on the sliding surface balances the entire weight of the body (including external forces) and the vertical forces on the contour.
3.  Total horizontal equilibrium is ensured when the horizontal component of the resistance forces acting on the sliding surface is in equilibrium with the horizontal forces acting on the contour.
4.  The equilibrium of the total moments is ensured when the sum of the moments of the differential forces between the slices is in equilibrium with the moments of the forces acting on the contour.

The relations used in determining the safety factor in the methods that satisfy the equilibrium of forces and moments (respectively, the GLE method, the generalized limit equilibrium method, Morgenstern–Price) are as follows:

The equation of the safety factor that satisfies the equilibrium of moments is

$$F_m = \frac{\sum(c'\beta + (N - u\beta)R\,tg\varnothing')}{\sum Wx - \sum Nf \pm \sum Dd}. \tag{1}$$

The equation of the safety factor that satisfies the equilibrium of forces is

$$F_m = \frac{\sum(c'\beta cos\alpha + (N - u\beta)R\,tg\varnothing' cos\alpha)}{\sum Nsin\alpha - \sum Dcos\omega}, \tag{2}$$

where $c'$—effective cohesion; $\varnothing'$—effective angle of internal friction; $u$—pore pressure; $N$—normal force at the base of the slice; $W$—slice weight; $D$—linear loading; $\beta$, $R$, $x$, $f$, $d$, $\omega$—geometrical parameters; $\alpha$—inclination of the slice base.

Lambda represents the ratio between the two equations that represent the safety factors and the equilibrium of moments or forces. Depending on this ratio, it is possible to describe graphically, for each method of calculation, the degree of total or partial satisfaction of the static balance. This is exemplified by the graphs presented in Figures 9 and 10.

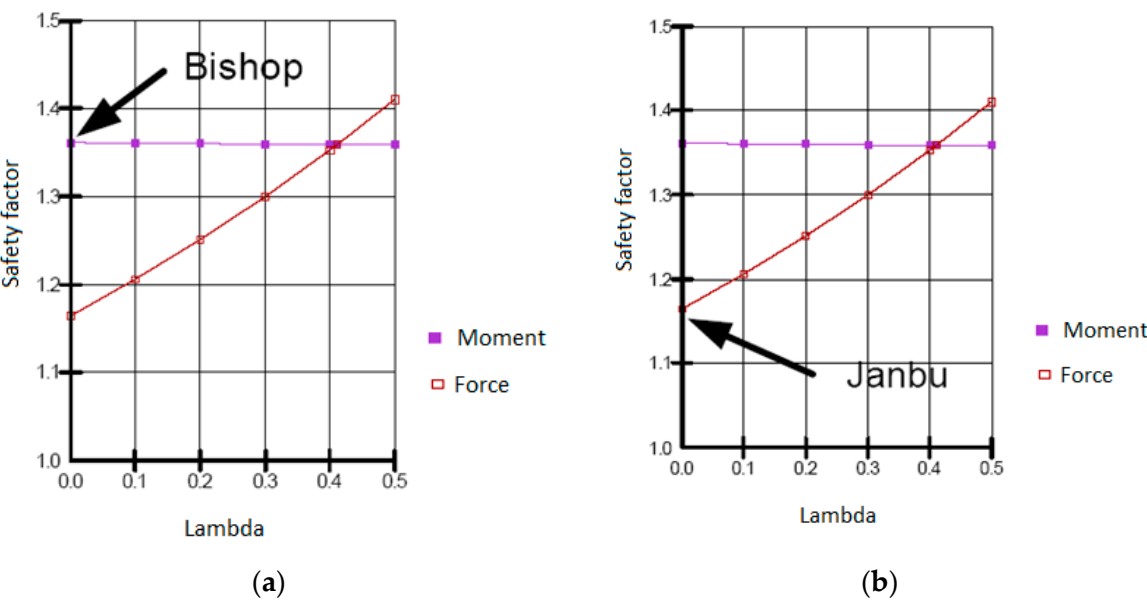

**(a)**　　　　　　　　　　　　　　　**(b)**

**Figure 9.** Static equilibrium: (**a**) by the Bishop method; (**b**) by the Janbu method.

The stability analyses were carried out with a software specialized for geotechnical problems (SLOPE software—GeoStru 365, produced by the GeoStru Company, Cluj-Napoca, Romania), which allows the analysis of the stability of the potential sliding surfaces using a group of limit equilibrium analysis methods with the division of the section into vertical slices.

The analysis methods can be used for individual surfaces that have been identified in situ as potential critical slip surfaces, or several slip surfaces can be generated by the method of the polygon of the centers of the surfaces by generating circular surfaces on certain segments restricted to the slope geometry by imposing in depth some limits, discontinuities, etc.

In this case, the method of generating the sliding surfaces was utilized by using a polygon that represents the geometric location of the centers of the sliding surfaces in a matrix with the resolution given by the incrementing step of the network. The radius of the

surfaces varies between the minimum and maximum values with a given step of increase that allows the most uniform coverage of the potential sliding surfaces (Figure 11).

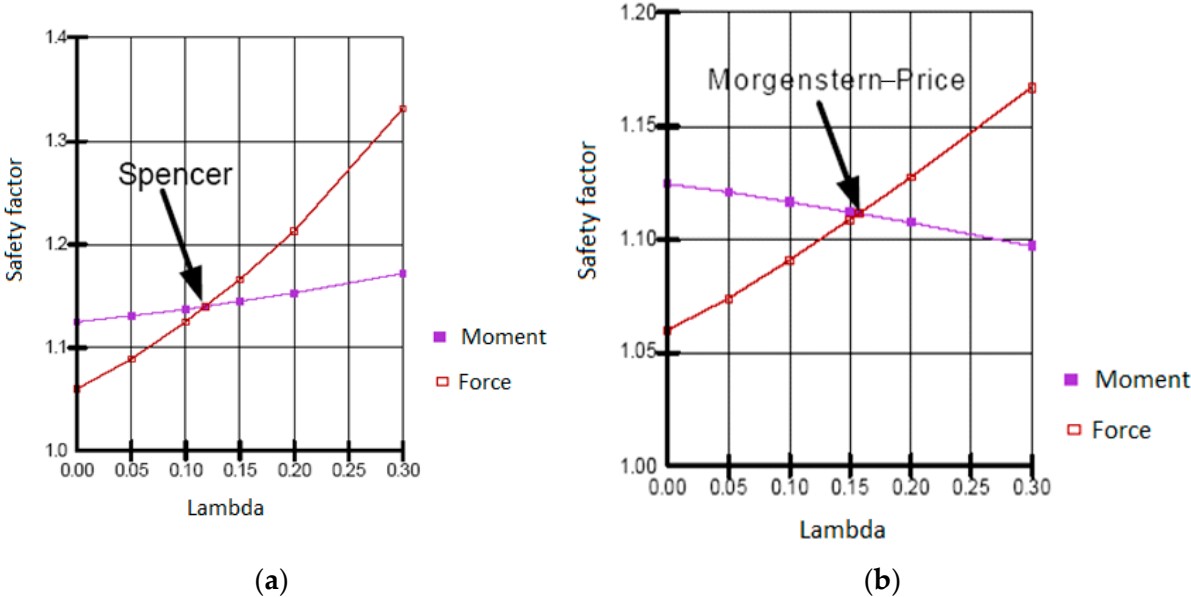

(a)      (b)

**Figure 10.** Static equilibrium: (**a**) by the Spencer method; (**b**) by the Morgenstern–Price method.

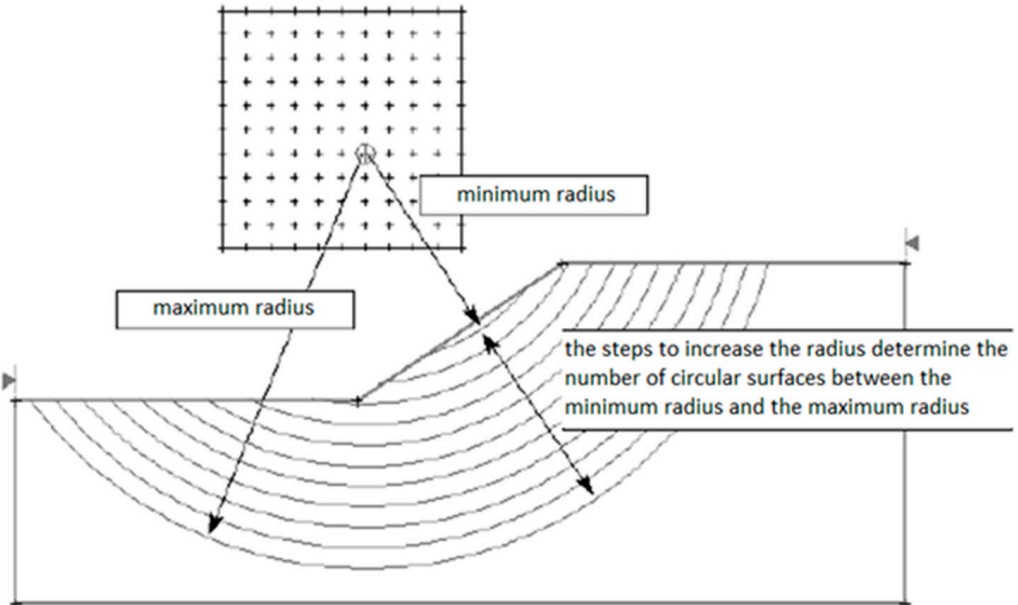

**Figure 11.** Radius of potential sliding surfaces.

The known hydrostatic level is introduced into the model through the form of the depression curve, and if it is not known, it can be simulated through the pore pressure network determined from the infiltration analysis through the dam body, as can be seen in Figures 12 and 13.

The shear parameters (friction angle and cohesion) introduced for each material included in the calculation section can be isotropic or anisotropic, respecting several fracture criteria, namely Mohr–Coulomb (in the present analyses), Hoek–Brown, Barton–Gangsters, etc.

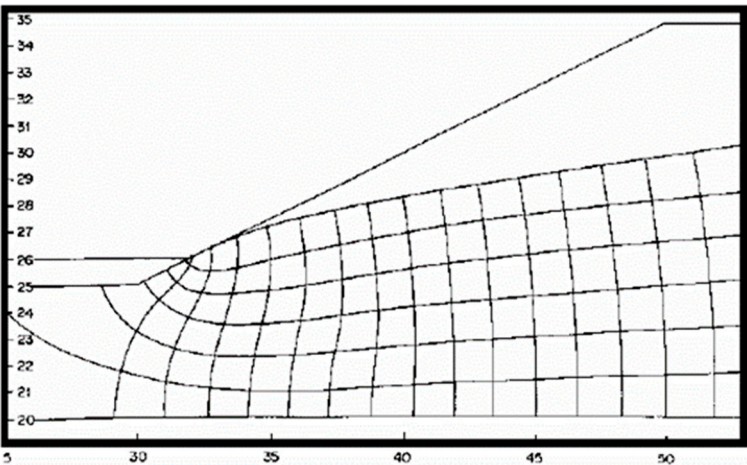

**Figure 12.** Pore pressure network determined by infiltration analysis.

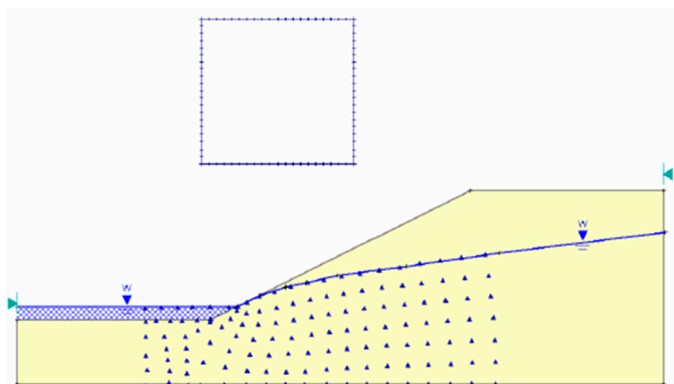

**Figure 13.** Introduction of the hydrostatic factor through the pore pressure network determined from the infiltration calculation.

### 3.2. In Situ and Laboratory Studies

The three compartments of the Gura Roşiei tailings dam served the preparation plant located about 1 km downstream from them. The oldest compartment, namely No. 1, has been out of function for about 37 years.

Compartment No. 1 was initially a triangular-shaped plateau pond, closed on three sides with a priming dike over which the settled material was elevated. Starting in 1966, with the construction of compartment 2, it elevated above the southern side of compartment 1, covering it with settled material. The height of compartment 1 is approximately 15 m. It is bordered to the east by the Abrudel stream, to the west by the slope of the Dăroaia hill, and to the south by compartment No. 2. Due to the fact that the tailings dam has been out of operation for almost 37 years, the deposited material has consolidated and drained. The slopes and the beach are dry, although there is a rather heavily eroded area towards the outer part of the western slope. The land at the foundation of the pond is mainly made up of coarse alluvial deposits that constitute the terrace of the Abrudel stream.

Compartment No. 2 of the Gura Roşiei tailings dam is a coastal pond, bordering to the east with the Abrudel stream and with compartment No. 3, in the north partially resting on compartment No. 1, and in the west supported by the slope of the Dăroaia hill. The slopes of compartment 2 are dry, being made of predominantly medium–fine sandy material (from a particle size point of view) and without suffusion phenomena. The land on which compartment 2 is located is made up of coarse terrace deposits and diluvial clay deposits towards the Dăroaia hill. Due to the non-functionality of the guard ditches, the rainwater flows directly over the slopes, causing erosional phenomena highlighted by

quite significant ditches and ravines. Compartment 2 is over 45 meters high and has been decommissioned for approximately 22 years.

Compartment No. 3 came into operation in 1974 and has a height of about 42 meters. The western side rests on the slope of the Dăroaia hill, which is made up of clayey diluvial deposits that do not allow water to drain. On the northern side, compartment 3 rests on compartment 2. At the base are the terrace deposits of the Abrudel stream, which allowed good drainage of the waters from the body of the pond. In the last stage of activity (from July 1988), compartment 3 operated alternately with Valea Selistei for short periods of time (1–2 months) in order not to allow the piezometric level in the body of the pond to rise to a level that could have affected the stability in general (a fact highlighted by the calculations performed in the respective period).

The works on which this study was based consisted of field observations, geotechnical drillings (five in compartment 1; two in compartment 2, and three in compartment 3), and physical–mechanical analyses of the collected samples. In addition, the data analyzed at the level of studies from 1988 were also taken into consideration in order to have a broader view of the evolution of the representative geomechanical parameters [32–36]. Due to the relatively small depth, the drilling could not highlight a hydrostatic level, except for the areas with excess humidity, which are represented by certain lenticular intercalations of sandy dust that yield water very hard. These areas are not continuous and cannot define a reference hydrostatic level.

*3.3. Calculation of Seepage in the Dikes' Body of the Dams and through the Foundation Using the Finite Element Method*

The infiltrations that take place through the retention elements of the hydrotechnical works, respectively, through the dikes of the tailings dams in the mining industry, can be numerically modeled, finally determining the hydrodynamic spectrum of the flow and the specific unit flow rates for the considered section. The determination of the hydrodynamic spectrum is based on Laplace's continuity theory, considering the hypothesis of steady-state flow. The hydrodynamic spectrum is known by determining the current and equipotential lines of the flow through the respective medium. In order to construct the hydrodynamic spectrum, the boundary conditions are established at the level of the studied location (known hydrostatic levels, impermeable limits, piezometric values at certain points, etc.). After establishing the boundary conditions, current lines and equipotential lines are constructed through successive iterations so that all elements in the spectrum have the same ratio between length and width. After constructing the hydrodynamic spectrum, the exfiltration flows are calculated on the respective section using the formula

$$q = kh_{max}\frac{N_f}{N_d} \times n, \tag{3}$$

where $N_f$—number of current lines; $N_d$—number of equipotential lines; $n$—ratio between the length and width of the network (spectrum) elements; $h_{max}$—difference between the upstream and downstream water level; and $k$ = permeability coefficient.

**4. Risk Analysis**

For the situation of the Gura Roşiei tailings dam, a risk analysis was performed before the stabilization and greening works were carried out. In carrying out the risk analysis study, reference points usable at different levels are necessary [57–59]. It must be accepted that the risk cannot be reduced to zero; therefore, the limit of the risk that can be borne by people in their current activities appears as the most important value. The qualitative analysis has as its main objective the establishment of the list of possible hazards; it makes it possible to rank the events in order of risk and presents the first step in the methodology of performing the quantitative risk analysis. There are two large categories of techniques within which a series of general components can be distinguished (Table 3) [60–62]:

- Qualitative analyses are used to identify hazards (Hazard and Operability Study—HAZOP). The qualitative analysis has as its main objective the establishment of the list of possible hazards; it makes it possible to rank the events in order of risk and presents the first step in the methodology of performing the quantitative risk analysis.
- Quantitative analyses are used to assess hazards to decide how to act in order to eliminate or reduce the risk (Hazard Analysis—HAZAN).

**Table 3.** The difference between the HAZOP technique and the HAZAN technique.

| HAZOP Technique | HAZAN Technique |
|---|---|
| Identify hazards | Evaluate the hazards |
| It is preferred for use in every project | It is a selective technique used mainly for systems potentially exposed to major accidents |
| Qualitative technique | Quantitative technique |
| It is carried out by a team | It is made by one or two experts in the field |
| It is also called "But What If?" | |

The order of application is from qualitative identification to quantitative analysis.

The qualitative determination of the consequences is carried out by classifying them into five levels of severity, an internationally accepted methodology used in risk assessment studies (Table 4).

**Table 4.** Risk severity levels [32,36].

| No. | Risk Severity Level | Effects |
|---|---|---|
| 1 | Insignificant | For humans (the population): insignificant injuries. Ecosystems: some minor adverse effects on a few species or parts of the ecosystem, short-term and reversible. Socio-political: insignificant social effects without cause for concern for the community. |
| 2 | Minor | For humans (population): insignificant injuries. Ecosystems: some minor adverse effects on a few species or parts of the ecosystem, short-term and reversible. Socio-political: insignificant social effects without cause for concern for the community. |
| 3 | Moderate | For the population: medical certificates are required. Economic: reduction of production capacity. Emissions: emissions within the objective can be contained with external help. Ecosystems: temporary and reversible damage; damage to habitats and migration of animal populations; plants unable to survive; air quality affected by compounds with potential long-term health risks; possible damage to aquatic life; pollution requiring physical treatments; limited contamination of the soil which can be remedied quickly. Socio-political: Social effects of moderate concern for the community. |
| 4 | Major | For people (population): special injuries. Economic: interruption of production activity. Emissions: off-site emissions without harmful effects. Ecosystems: the death of some animals; large-scale damage; damage to local species; and the destruction of extensive habitats; air quality requiring "safe refuge" or the decision to evacuate; soil remediation only possible through long-term programs. Socio-political: Social effects of serious concern for the community. |
| 5 | Catastrophic | For people (population): death. Economic: stopping the production activity. Emissions: off-site toxic emissions with harmful effects. Ecosystems: the death of animals in large numbers; the destruction of flora species; the quality of the air requiring evacuation; permanent contamination of extensive areas of the soil. Socio-political: social effects of particular concern to the community. |

The analysis of the probability of the occurrence of risks is carried out by classifying them in the five levels, internationally accepted and used in different versions (Table 5) [48–53].

**Table 5.** Levels of risk occurrence probability [34,36].

| No. | Probability | When It Can Be Produced | Frequency (Probability) of Occurrence |
|---|---|---|---|
| 1 | Rare | Only in exceptional conditions | $10^{-12}$ ani |
| 2 | Not probably | It could happen sometime | $10^{-8}$–$10^{-12}$ ani |
| 3 | Posible | It can happen sometime | $10^{-6}$–$10^{-8}$ ani |
| 4 | Probably | It can happen in most situations | $10^{-4}$–$10^{-6}$ ani |
| 5 | Almost certainly | It is expected to happen in most situations | $<10^{-4}$ ani |

For the assessment of the risk, the level of risk is determined as the product between the level of gravity (consequence) and the level of probability of the analyzed event. The results obtained from this analysis, for the situation of the Gura Roşiei tailings dam, were transposed into the risk matrix in Table 6.

**Table 6.** Qualitative risk assessment matrix and risk levels.

| | | | SEVERITY | | | | |
|---|---|---|---|---|---|---|---|
| | | | Negligible | Medium | Grave | Very Grave | Catastrophic |
| | | | 1 | 2 | 3 | 4 | 5 |
| PROBABILITY | Very frequently | 1 | 1 | 2 | 3 | 4 | 5 |
| | Frequently | 2 | 2 | 4 | 6 | 8 | 10 |
| | Rare | 3 | 3 | 6 | 9 | 12 | 15 |
| | Extremely rare | 4 | 4 | 8 | 12 | 16 | 20 |
| | Improbable | 5 | 5 | 10 | 15 | 20 | 25 |

Existing measures or those that will need to be implemented in order to have an appropriate level of safety are determined according to frequencies and consequences. In order to assess the potential mining risks associated with the conditions of the Gura Roşiei tailings dam, numerical values were assigned for each probability level of potential risk manifestation and for each level of severity. The thick black line in the potential risk matrix is the extent to which the probability of these identified risks should be maintained; in these situations, procedures and work processes must be followed, which means that no additional risk reduction measures would be needed. In the yellow zone, the risks could be reduced to the lowest level considered tolerable, but this reduction involves identifying and implementing the necessary technical or organizational measures. For all risks with frequencies in the red zone, the immediate implementation of technical measures is required, using all available resources to reduce the level of risk(s).

## 5. Results and Discussion

### 5.1. Stability Study Results

In this study, seven calculation sections were checked, whose directions and locations are as follows:

- Compartment 1: Section I-I′ S-N; Section II-II′ W-E;
- Compartment 2: Section III-III′ NE-SW; Section IV-IV′ NW-SE;
- Compartment 3: Section V-V′ NE-SW; Section VI-VI′ E-W; Section VII-VII′ SE-NW.

For sections I and II, two situations were considered: the initial one corresponding to the current one and another in which a rise of 1.5 m in the beach and 2 m in the support dike was simulated. In addition, for each individual situation, two hypotheses were considered, static and pseudo-static, in which the seismic stresses intervene at the magnitude of the site area according to the norm P100/1992 (seismic zone F and seismic coefficient Ks = 0.08 g).

All situations were analyzed by four different methods that satisfy the static equilibrium of forces or moments (Bishop and Janbu) or simultaneously of forces and moments (Spencer and Morgenstern–Price).

From the point of view of the capacity to absorb the efforts given by the 1.5–2.0 m elevation of compartments 1 and 2, the laboratory data from oedometer tests on samples collected in the field reveal the corresponding parameters for the deformation modulus ($M_{2-3}$ = 143–250 daN/cm$^2$ and the settlement coefficient $e_p$ = 13–23 mm/m). The geotechnical parameters used in the calculations are presented in Table 7.

**Table 7.** Input geotechnical parameters of the materials in the tailings dam body used in the calculations.

| No. | Material Type | Volumetric Weight (kN/m$^3$) | Internal Friction Angle (Degree) | Cohesion (kPa) |
|---|---|---|---|---|
| 1 | Fine–medium sandy pond material | 15.79 | 33 | 10 |
| 2 | Dusty sand pond material | 15.79 | 30 | 20 |
| 3 | Priming dike | 19.62 | 35 | 0 |
| 4 | Terrace deposit | 17.66 | 35 | 10 |
| 5 | Diluvial deposit | 18.00 | 20 | 30 [1] |

[1] values obtained in the Geotechnical laboratory of the University of Petrosani.

For the sections where the stability was checked in the case of a 1.5 m elevation of the tailings deposit, the hydrostatic level was assimilated to the maximum that was used in the case of the operation of the two compartments. The same was conducted in the case of the other modeled profiles because the non-functioning of the draining elements could not ensure the corresponding decrease in the depression curve. The results of the calculations are presented in Figures 14–19 and Table 8.

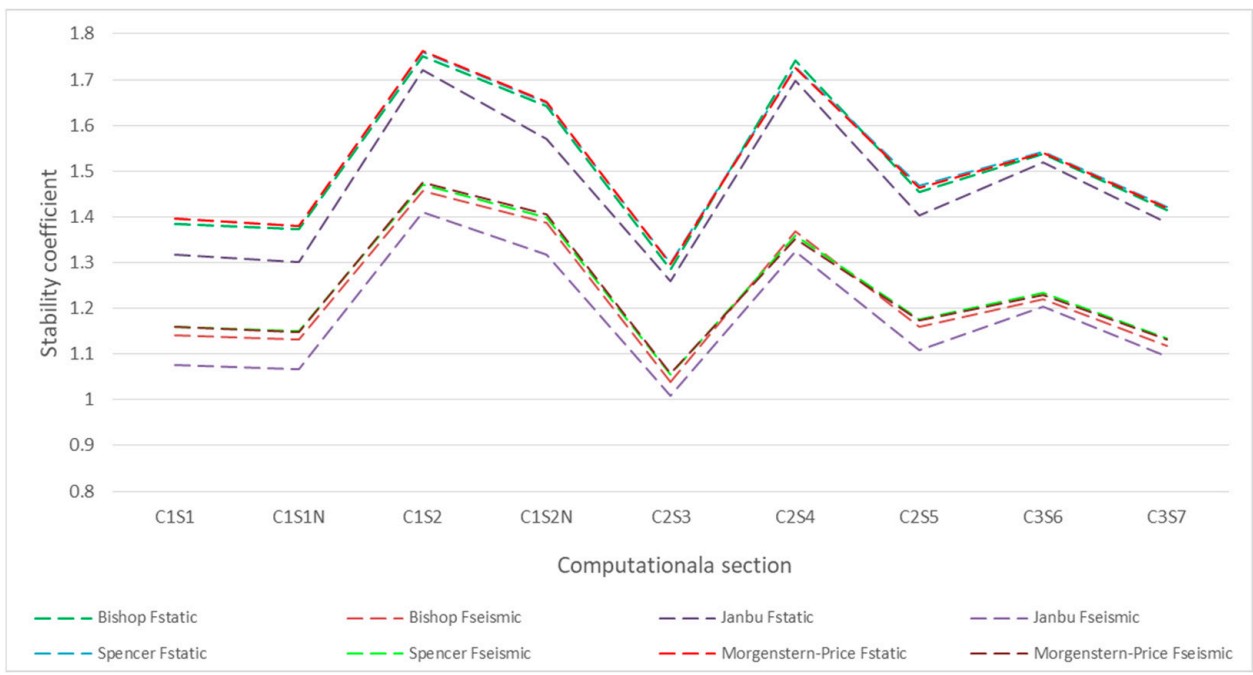

**Figure 14.** Variation of the stability coefficient static and seismic in the three compartments of the tailings dam.

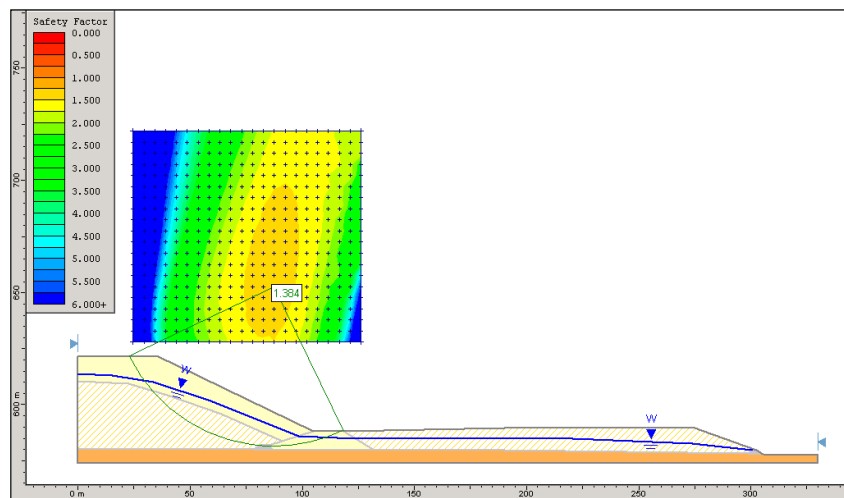

**Figure 15.** Static analysis of the stability of compartment 2 of the Gura Roşiei dam (with support on compartment 1).

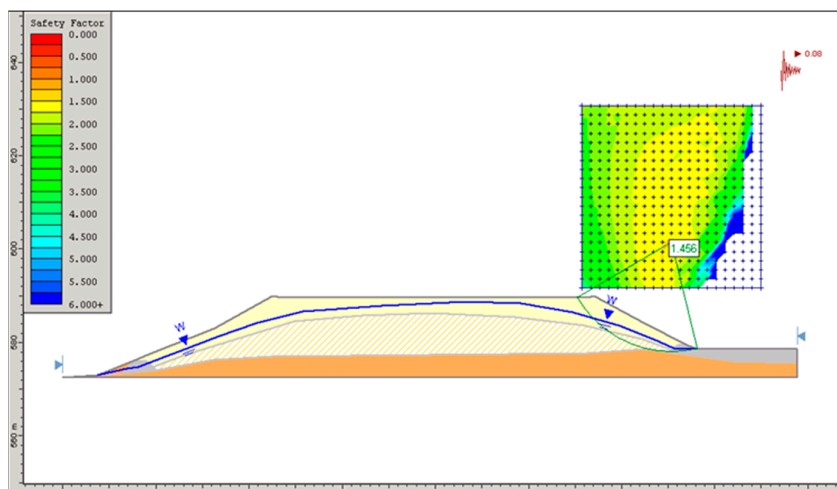

**Figure 16.** Seismic analysis of the stability of compartment 1 of the Gura Roşiei dam.

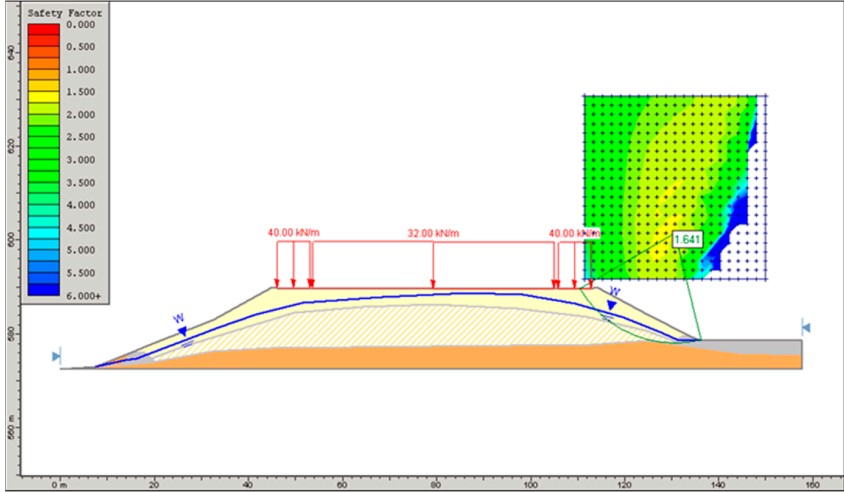

**Figure 17.** Static analysis of the stability of compartment 1 of the Gura Roşiei dam (elevation with 1.5 m of sterile deposition).

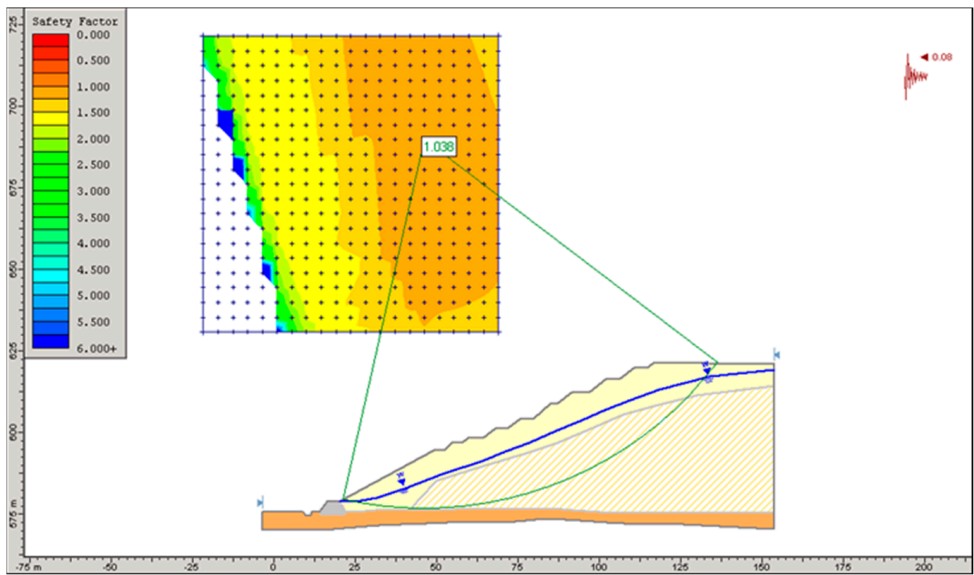

**Figure 18.** Seismic analysis of the stability of compartment 2 of the Gura Roşiei dam (Profile 3-3′).

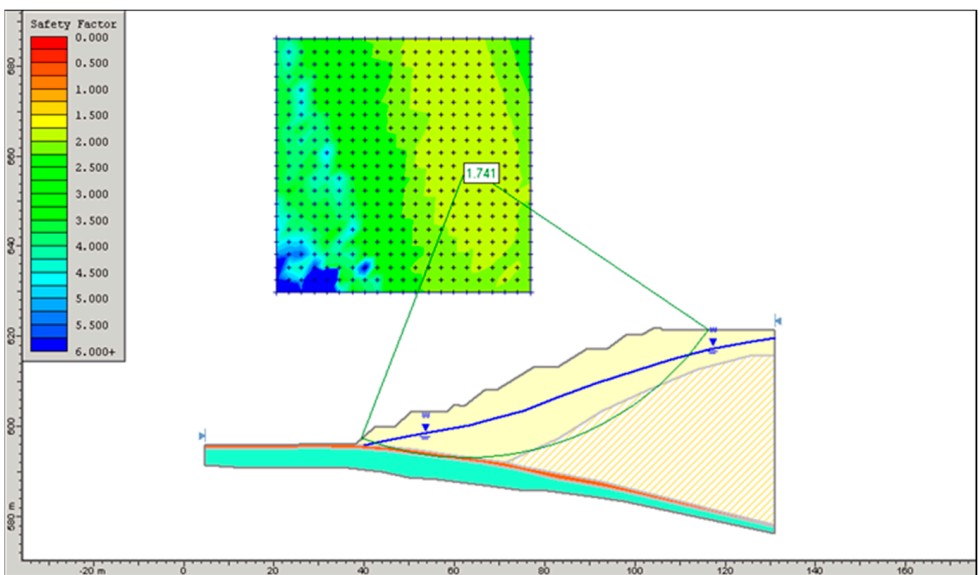

**Figure 19.** Static analysis of the stability of compartment 2 of the Gura Roşiei dam (Profile 4-4′).

**Table 8.** Values of the stability coefficient in the considered sections.

| Computational Section | Computing Method [1] | | | | | | | |
|---|---|---|---|---|---|---|---|---|
| | Bishop | | Janbu | | Spencer | | Morgenstern–Price | |
| | $F_{static}$ | $F_{seismic}$ | $F_{static}$ | $F_{seismic}$ | $F_{static}$ | $F_{seismic}$ | $F_{static}$ | $F_{seismic}$ |
| Compartment 1, Section 1 (C1S1) | 1.384 | 1.140 | 1.317 | 1.075 | 1.397 | 1.160 | 1.397 | 1.159 |
| Compartment 1, Section 1 new (C1S1N) | 1.372 | 1.132 | 1.302 | 1.067 | 1.381 | 1.151 | 1.380 | 1.149 |
| Compartment 1, Section 2 (C1S2) | 1.752 | 1.456 | 1.722 | 1.411 | 1.761 | 1.471 | 1.762 | 1.474 |

**Table 8.** *Cont.*

| Computational Section | Computing Method [1] | | | | | | | |
|---|---|---|---|---|---|---|---|---|
| | Bishop | | Janbu | | Spencer | | Morgenstern–Price | |
| | $F_{static}$ | $F_{seismic}$ | $F_{static}$ | $F_{seismic}$ | $F_{static}$ | $F_{seismic}$ | $F_{static}$ | $F_{seismic}$ |
| Compartment 1, Section 2 new (C1S2N) | 1.641 | 1.386 | 1.570 | 1.317 | 1.648 | 1.399 | 1.652 | 1.405 |
| Compartment 2, Section 3 (C2S3) | 1.284 | 1.038 | 1.260 | 1.008 | 1.298 | 1.056 | 1.296 | 1.057 |
| Compartment 2, Section 4 (C2S4) | 1.741 | 1.369 | 1.698 | 1.325 | 1.729 | 1.359 | 1.725 | 1.352 |
| Compartment 3, Section 5 (C3S5) | 1.455 | 1.159 | 1.403 | 1.108 | 1.468 | 1.177 | 1.464 | 1.173 |
| Compartment 3, Section 6 (C3S6) | 1.537 | 1.219 | 1.519 | 1.203 | 1.542 | 1.234 | 1.540 | 1.230 |
| Compartment 3, Section 7 (C3S7) | 1.414 | 1.119 | 1.388 | 1.094 | 1.422 | 1.135 | 1.419 | 1.132 |

[1] SLOPE software produced by the GeoStru Company.

It can be observed that in the case of an elevation of 1.5 meters, the safety factor decreases very little, but its value, especially for compartment 2, calls for an alternative exploitation of it (with compartment 1) for a short period of time (maximum 2 months). In addition, the north-eastern slope of compartment 2 requires special attention in the case of raising the hydrostatic level or a hydraulic screen too close to the crest.

*5.2. Infiltration Modeling*

The numerical modelling of infiltrations using the finite element method was realized for two two-dimensional sections identical to those in the stability analysis, representing the current situation. The pond was divided into three main blocks: the dike, the body of the pond made of deposited material, and the foundation layer. The filtering properties taken into account were the following:

1. The terrace deposit (clogged up in the digging area), $K = 2.90 \times 10^{-5}$ m/s;
2. The priming dike, $K = 15.80 \times 10^{-5}$ m/s;
3. Dam material (medium–fine sand), $K = 1.16 \times 10^{-5}$ m/s;
4. Dam material (dusty sand), $K = 5.80 \times 10^{-5}$ m/s.

From the point of view of permeability coefficients, it was considered that the medium studied is isotropic, so $k_x = k_z$. The modeling was carried out under the assumption of a free flow in a steady state due to the fact that the piezometric level is not known a priori (this being one of the borders). The calculation sections were discretized by triangular finite elements with six nodes. The hydraulic conductivities were assigned to each individual block, following a complex Fredlund and Xing function type for the materials in the modeled section (Figures 20–23). These distribution laws refer to how the pressure of the suction matrix varies with the permeability of the material. The first point on the ordinate represents the saturated permeability of the material (suction pressure is equal to 0).

The boundary conditions imposed the assignment of nodes from the "upstream" end, from the point of view of the depression curve of the section, to a hydrostatic pressure equal to the height of the water column at the level existing during operation. At the opposite end, at the level of the foot of the slope (Compartment 1) and its external contour, the corresponding nodes were marked as a border with unknown initial conditions, the program allowing the determination of the point of convergence with the outer slope (towards the priming dike). After running the software through successive iterations with an appropriate convergence error ($10^{-6}$), the depression curve, the hydrodynamic spectrum, and the infiltration rate were determined for two identical flow sections with the

profile passing through compartments 2 and 1 in the south–north direction. The unit flow rates of infiltrations through the dike are presented in Table 9.

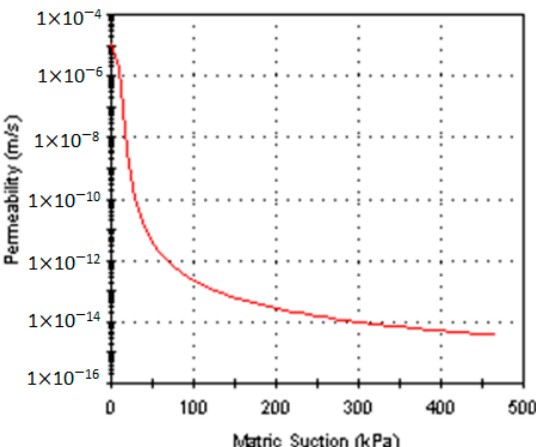

**Figure 20.** Fredlund and Xing function, medium−fine sand.

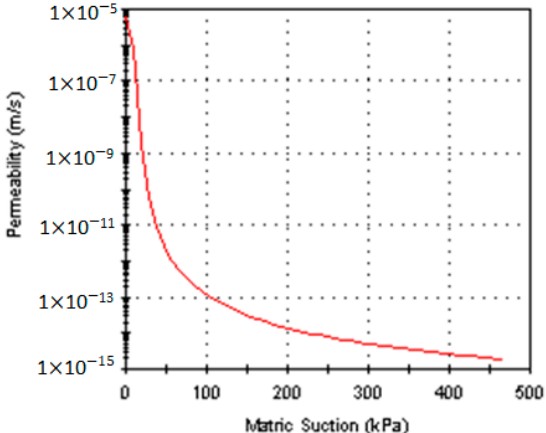

**Figure 21.** Fredlund and Xing function, fine dusty sand.

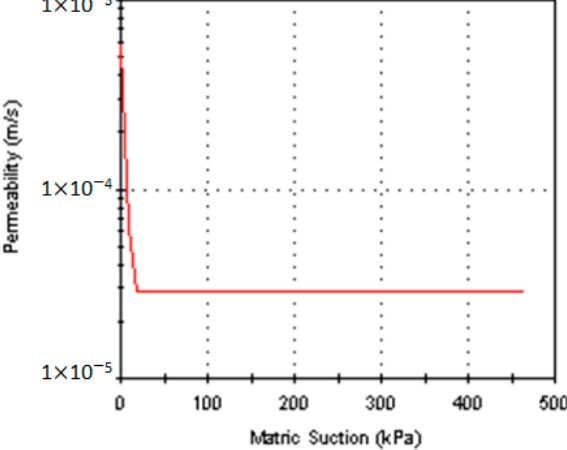

**Figure 22.** Simple function, priming dike.

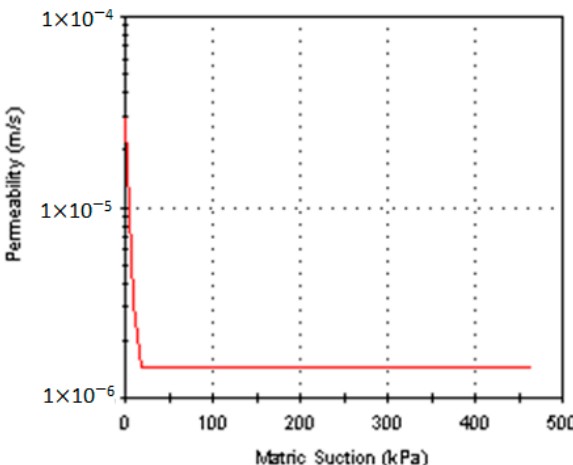

**Figure 23.** Simple function, terrace deposit.

**Table 9.** Calculated values of seepage through the dike.

| Calculation Section | Infiltrations through the Dike | |
| --- | --- | --- |
| | **(m³/s)** | **(m³/day)** |
| Initial Section 1 | $2.221 \times 10^{-7}$ | 0.020 |
| Section 1 superelevation | $4.324 \times 10^{-6}$ | 0.374 |

Analyzing the results of the stability study, the calculation of infiltrations, and the conclusions of the studies regarding the behavior during exploitation, a series of conclusions emerge regarding both the current state of ponds No. 1, 2, and 3 of Gura Roşiei and also the conditions of a possible future exploitation.

Thus, the hydrogeotechnical study carried out on the site of the tailings dams aimed to evaluate the degree of stability of the three ponds in order to specify whether the idea of raising them by 1.5–2 m with tailings is feasible. From the point of view of the stability calculations performed in the hypothesis in which the three ponds become active for the storage of tailings, assuming a corresponding piezometric level, the resulting safety factors are relatively close to the standard values (Fs ≥ 1.4) for the static analysis, and in seismic conditions, they are at the limit of equilibrium. The NE slope of dam No. 2 (Section 3.3) shows values below the standard safety limit for this type of work. In addition, settling pond No. 3 presented from the calculations as being totally inadequate for the elevation.

Due to the reduced capacity to yield water from its pores, the settled material is still in a saturated state (the exfiltrated flow rates being small), and it is assumed that the foundation land, made up of the terrace deposits of the Abrudel stream, is clogged at the interface with the settled material, and we can no longer rely on it to naturally drain the excess moisture from the body of the dam. If the solution of raising these dams is adopted, it will be necessary to provide for the important drainage of the waters that flow into the tailings deposits (tailings water and precipitation water). From the stability analysis of the Gura Roşiei tailings dam, the safety coefficients presented in Table 10 were obtained.

**Table 10.** Safety coefficients obtained for the Gura Roşiei tailings dam.

| Tailings Dam Compartment | Safety Coefficient [1] on: | |
| --- | --- | --- |
| | **Static Load** | **Dynamic Load** |
| Compartment No. 1 | 1.31–1.76 | 1.07–1.47 |
| Compartment No. 2 | 1.24–1.72 | 1.008–1.35 |
| Compartment No. 3 | 1.39–1.54 | 1.09–1.23 |

[1] obtained with the SLOPE software.

Consequently, the site formed by the three analyzed tailings dams from Gura Roşiei presents major disadvantages for being used for future storage of the preparation waste, namely:

- Dam No. 3 cannot be used because it does not have the mechanical characteristics necessary for this type of hydrotechnical construction.
- Currently, there is only one clear water evacuation system (in dam number 2).
- Any elevation of these dams requires a corresponding occupation of the land surface on the coast on which the deposit rests, namely, on privately owned land where expropriation may lead to negotiations with uncertain results.
- Due to the weak geotechnical characteristics of the land on which a new dam would be located (represented by the three existing dams), particularly expensive work will be necessary, as well as taking a long time to complete.

According to the EU provisions not to allow after 31 December 2006 the storage of liquid waste in non-waterproof locations, a geomembrane waterproofing of the entire surface of ponds 1, 2, and 3 will be mandatory.

Based on the performed studies, it is recommended to design and carry out safety and greening works in order to avoid pollution of the surrounding environment and, at the same time, restore its location in the natural circuit of the area. It is not recommended to build a new storage space for the preparation tailings on the site of No. 1, 2, and 3 Gura Roşiei dams.

## 6. Technical Proposals

Due to the weather conditions (rain and wind) and the lack of maintenance work, the access roads to and on the Gura Roşiei tailings dam are damaged, making access difficult or even impossible in some areas. For the access of machines and the transport of materials to the working areas, it is necessary to build a technological road network around the dam. A solution for greening the dam in its dry version without the presence of water accumulations on its surface can be provided. This can be achieved by combining two types of works: namely, by constructing a temporary channel (with a length of 200 m) that will drain the clarified water and that will deepen as the water level drops, so that the current water does not entrain the deposited tailings. The water from the hydraulic screen will discharge into the guard channel dug provisionally in the natural terrain and which discharges into the Dăroaia stream. Since the depth of the accumulation is maximum around the cleared water drainage system with reverse probe, and in order not to deepen the temporary drainage channel too much, the metal sections that make up the reverse probe will be dismantled, as the water level drops. When reaching the bottom of the reservoir, a submersible pump will be used to completely evacuate the water from the dam platform. After draining the water from the surface of the dam, a period of time is necessary for the land to dry. The existing water drainage system (reverse well) will be kept in operation, as a backup and emergency system in case of heavy rainfall, until the completion of the greening works, when it will be consolidated, by filling with concrete along its entire length.

Through the systematization works, the dam will be given a new shape and with a stable slope at a slope of 1:3, and a platform will be created on the upper surface of the dam with a slope of 1% downstream. The tailings volume from the upper part of the dam, compartments No. 2 and 3, will be excavated to depths h = (0–5) m, and it will be transported and leveled on the surface of compartment No. 1, bringing all the compartments to the same level. Upon completion of the systematization and reprofiling works, four boreholes with a length of 35–40 m and equipped with piezometric tubes for monitoring the hydrostatic level in the body of the dam will be made on the dam platform (approximately 15 m from the edge of the slope).

In order to ensure the stability of the dam, in addition to reducing the angle of slope, a consolidation of the base of the dam slope will be carried out with rockfill (rough stone), which will also have the role of draining the water inside the pond or the water drained from the pond slope. This ballast prism must be built along the entire length at the base

of the dam for approximately 1350 m. The ballast prism will be built on the "reverse filter" system and will consist of a 0.5 m thick layer of ballast placed on the tailings slope of the dam and a surface layer of raw stone 1.5 m thick, having a height of 4 m. A 4 m wide bench will be provided at the upper part of the massif, which can be used for traffic during the post-closure monitoring of the pond, as well as in case of any interventions. The water collected from the drainage at the base of the stone massif will be discharged into the Abrudel River through three open channels, located along the ballast prism. A hydrometric gauge will be provided to monitor the flows discharged into the canal. In order to evacuate the water from the drains on the slopes, it is necessary to build two guard channels (coastal).

The closing and greening layers of the dam surface (slope and platform) may be composed of a 1 mm thick waterproof layer of geomembrane (draining geocomposite type TENAX TENDRAIN 750/2—geotextile/geogrid/geotextile); geocomposite drainage layer with minipipes; layer of sterile material of 35 cm; and a 15 cm thick layer of topsoil, fertilized and seeded with grass.

In order to collect and evacuate the water drained by the draining geocomposite layer on the surface of the slope, a 900 m long drain placed at the level of the berm of the ballast prism from the rockfill will be made. It is built when the waterproofing drainage layers (geomembrane and draining geocomposite) are made on the surface of the pond slope. For the platform, an INTERDRAIN GM 412 type draining geocomposite (geogrid/geotextile) will be installed over the geomembrane layer. Although this type of draining geocomposite has geotextile only on the upper part, due to the fact that the slope of the platform is 1% and that the greening layers are about 0.5 m thick, there is no need for an additional layer of geotextile for the protection of the geomembrane. The ballast layers of 0.35 m of coarse granular material and 0.15 m of topsoil will be built over the draining geocomposite and the greened surface of the platform will be seeded with a mixture of perennial plants specific to the pedoclimatic zone.

In order to monitor the hydrostatic level in the body of the dam, four piezometric tubes will be mounted on the dam platform, and to monitor the stability, of any horizontal or vertical movements, at the completion of the greening works, eight topographic markers must be installed as follows: four topographic markers on the dam platform, one next to each piezometric tube; three topographic markers on the slope of the dam; and a topographic landmark in natural terrain, near the guard channel, on the slope which supports the dam.

Through the safety and greening works of the Gura Roşiei tailings dam, the causes and sources of environmental pollution are not completely eliminated; a majority of them are eliminated, and others are reduced in intensity. Thus, the pond is encapsulated only on its free surfaces, eliminating the entrainment of fine particles in the form of dust under the action of wind gusts and the leakage of sterile material under the action of rains that can pollute the soil and waters around the site of the dam. At the same time, the amount of water infiltrated into the body of the dam is considerably reduced. This leads to increasing the stability of the pond and reducing the exfiltration of contaminated water from the body of the pond. However, the dam remains in contact with the natural terrain on an area of about 15 ha consisting of its base and the lateral surface that rests on the Dăroaia hill. Through these surfaces, a certain amount of water from the springs and the water table will seep into the dam, but at the same time, heavy metal pollution will occur in the upper part of the soil on these surfaces and in the water table in certain areas.

Considering the way in which the Gura Roşiei tailings dam was designed, built, and operated, as well as the works carried out for safety and greening, the following technical proposals can be formulated to be taken into account when designing, building, and closing some similar hydrotechnical constructions:

- carrying out a specialized study and an analysis of the physical–chemical composition of the material that makes up the deposit in order to accurately establish whether in its composition there are elements and useful mineral substances profitable from

an economic, strategic, and social point of view in order to recover them with the technologies existing at that time;

- during the design and construction of the tailings dam, it must be taken into account that there are surfaces of the deposit that will be in contact with the natural land and whose size differs depending on the type of dam. Thus, it is indicated that even from this phase, an isolation or waterproofing of this contact surface should be carried out, so as to avoid the penetration of water from the natural land into the body of the dam because this will contribute to the phenomenon of its instability, but also vice versa, to avoid infiltrating harmful and dangerous substances from the warehouse into the natural land, causing pollution of the groundwater in the area.

The greening and safety works of the tailings dam must be started in the first years of their operation so that the impact on the environment and human settlements is as low as possible, and the re-introduction of the occupied surfaces into the natural circuit is conducted as quickly as possible and under safe conditions.

## 7. Conclusions

The location formed by the three compartments that make up the Gura Roşiei tailings dam presents major disadvantages for a future storage of flotation tailings while being at the same time an imminent danger for the environment.

- From the point of view of the stability calculations performed in the hypothesis in which the three ponds become active for the storage of tailings, assuming a corresponding piezometric level, the resulting safety factors are relatively close to the standard values (Fs $\geq$ 1.4) for the static analysis, and in seismic conditions, they are at the limit of equilibrium. In some areas, the slope shows values below the standard safety limit for this type of work. The calculations showed that one of the compartments of the sedimentation pond is totally unsuitable for elevation. From the results obtained, it was found that the location formed by the three compartments that make up the Gura Roşiei settling pond presents major disadvantages for the future storage of the flotation tailings and is at the same time an imminent danger for the environment. A solution for greening the dam in its dry version without the presence of water accumulations on its surface can be provided.

**Author Contributions:** Literature review and analysis, M.T. and R.B.I.; methodology, M.T.; writing, M.T. and V.A.F.; experiments, M.T.; results analysis, M.T. and V.A.F. All authors have read and agreed to the published version of the manuscript.

**Funding:** This research received no external funding.

**Institutional Review Board Statement:** Not applicable.

**Informed Consent Statement:** Not applicable.

**Data Availability Statement:** Not applicable.

**Conflicts of Interest:** The authors declare no conflict of interest.

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
