# Peer review of "Stability Analysis of the Tailings Dam for the Purpose of Closing, Greening, and Ensuring Its Safety—Study Case"

_sustainability, doi:10.3390/su15097606_

Round 1

Reviewer 1 Report

This paper discusses the hydrogeotechnical and stability on the Gura RoÅŸiei tailing dam location. In summary, this paper is well organized. The content is good. I think it can be accepted. I only have few minor comment/suggestions to hopefully improve the quality of manuscript as below:

1. The innovation of the paper is not enough.

2. Check the figures. Not beautiful. In scientific journal, you must give very beautiful figures. i.e. the quality of Figure 13 needs to be improved.

3.Add a schematic diagram of the cross-section size of the tailings pond.

4.Provide a detailed information about the physical and mechanical properties of tailings.

5. Increasing the references in the past 5 years.

Author Response

List of Responses

Dear Editors and Reviewers:

Thank you for your letter and for the reviewers’ comments concerning the manuscript entitled “Stability analysis of the tailings dam for the purpose of closing, greening, and ensure its safety. Study case”. Those comments are all valuable and very helpful for revising and improving the paper, as well as providing important guidance for my research. We have studied the comments carefully and have made corrections, which we hope will meet your expectation. We would like to appreciate your insightful and constructive comments on our manuscript. We have fully revised and improved the contents of our paper according to your suggestions. The revised portions are marked in red in the paper. The main corrections in the paper and the responses to the reviewer’s comments are as following:

Responds to the reviewer’s comments:

Reviewer 1

This paper discusses the hydrogeotechnical and stability on the Gura RoÅŸiei tailing dam location. In summary, this paper is well organized. The content is good. I think it can be accepted. I only have few minor comment/suggestions to hopefully improve the quality of manuscript as below:

(1) The innovation of the paper is not enough.

Response: Esteemed reviewer, Thank you very much for your constructive suggestion.

The closing and greening of tailings dams is done taking into account the general stability of the pond and its related constructions, as well as the integration into the surrounding environment of the surfaces occupied by mining waste deposits (tailing dams). This study presents the results of the hydrogeotechnical and stability study carried out on the Gura RoÅŸiei tailing dam location. This analysis aims to evaluate the stability degree of the three compartments that make up the tailings deposit in order to carry out the closing and greening works of the tailing dam, and to conclude whether the idea of raising them by 1.5 ... 2 m with tailings is feasible. This study was based on field observations, geotechnical drilling, and physical-chemical analyses of the collected samples..

(2) Check the figures. Not beautiful. In scientific journal, you must give very beautiful figures. i.e. the quality of Figure 13 needs to be improved.

Response: Esteemed reviewer, Thank you very much for your constructive suggestion. According to the review comments, Figure 13 has been replaced (after revision, it became figure 14):

Figure 14. Variation of the stability coefficient static and seismic in the three compartments of the tailing dam.

(3) Add a schematic diagram of the cross-section size of the tailings pond.

Response: Esteemed reviewer, Thank you very much for your constructive suggestion.

The cross-section of the tailing dam was introduced in the content of this manuscript as follows:

Figure 1. Situation plan of the tailings dam.

In addition, we also present the situation plan considered to carry out this study:

 (4) Provide a detailed information about the physical and mechanical properties of tailings.

Response: Esteemed reviewer, Thank you very much for your constructive suggestion.

For the intended study, we determined the parameters used in the calculation software as input data. Samples were collected in the field, and the corresponding parameters for the deformation modulus were determined in the laboratory from the oedometer tests (M2–3 = 143–250 daN/cm2 and the settlement coefficient ep = 13–23 mm/m). The geotechnical parameters used in the calculations are presented in Table 7 (Input geotechnical parameters of the materials in the tailings dam body used in the calculations). Please kindly see page 17.

(5) Increasing the references in the past 5 years.

Response: Esteemed reviewer, Thank you very much for your constructive suggestion. It is true that the bibliography includes publications of great value, even if they have not been published. Of special value are the publications that appeared in the last decades and also in the last few years. We do not claim to have exhausted the studied bibliographic sources; on the one hand, we studied the older literature, but we tried to accumulate as much information as possible from the publications of recent years. We considered that the information found in the articles introduced after the revision of the manuscript is also of particular value; these are marked in red:

Johari, A., S. Mousavi, S., Hooshmand Nejad, A. A seismic slope stability probabilistic model based on Bishop's method using analytical approach. Scientia Iranica A, 22(3), 728-741. 2015.

Johari, A., Khodaparast, A.R. Analytical stochastic analysis of seismic stability of infinite slope Soil Dynamics and Earthquake Engineering, 79, Part A, Pages 17-21. 2015.

Johari, A., Rahmati, H. System reliability analysis of slopes based on the method of slices using sequential compounding method. Computers and Geotechnics, 114, 103116. 2019.

Sincerely yours,

Mihaela Toderas on behalf of all the authors

Reviewer 2 Report

Thanks for this opportunity to review this manuscript. I appreciate the author's contribution to the field and believe that the paper is valuable and provides new insights into the subject matter.

This manuscript mainly talking about three zone of Gura Rosiei tailing dams and itself mechanical property and hydrogeology element which decide the influence of environment.

The tailing dam are the Special structures, and the mechanical property are complicated. This manuscript utilizing the Fellenius, Taylor, Bishop, Morgenstern-Price, Spencer, Janbu, and Finite Element Method to solve and analyze the stability and water-proof performance. Therefore, to draw the conclusion of stability, risk occurrence probability and eco-friendly redevelopment.

The manuscript is well-structured, well-written, and the experimental design is sound. However, there are a few areas that could be improved:

1.     The results and the effort by authors didn’t embody in the abstract. The reader couldn’t catch the key content fast and accurately;

2.     The Introduction in chapter 1 didn’t show the vertical view and made readers hard to get the environment, height, the direction of river of the tailing dam. And can’t to know the influence of pollution made by heavy metal etc.;

3.     Three zone of tailing dam didn’t show clearly by the chart and hard to get the configuration of each part of tailing dam. Such as drainage, pollutant filtration equipment, etc. In the same time, the detail of tailing dam in manuscription are clearly. Therefore, should to illustrate the method how to get the details of dimension of the dam.

4.     The conclusion manuscript given hardly to share in another project and which tailing dam can fit this conclusion didn’t illustrate.

Therefore, this is a valuable manuscript that requires further improvement. I hope the authors will carefully consider my suggestions and pay more attention to the rigor of methods, depth of argumentation, and reliability of results in future studies.

Thank you again to the editor for giving me the opportunity to review this paper. I look forward to seeing the authors achieve even more outstanding results in their future research.

Congratulations on a job well done.

Best regards,

Liuhua Yang

Author Response

List of Responses

Dear Editors and Reviewers:

Thank you for your letter and for the reviewers’ comments concerning the manuscript entitled “Stability analysis of the tailings dam for the purpose of closing, greening, and ensure its safety. Study case”. Those comments are all valuable and very helpful for revising and improving the paper, as well as providing important guidance for my research. We have studied the comments carefully and have made corrections, which we hope will meet your expectation. We would like to appreciate your insightful and constructive comments on our manuscript. We have fully revised and improved the contents of our paper according to your suggestions. The revised portions are marked in red in the paper. The main corrections in the paper and the responses to the reviewer’s comments are as following:

Responds to the reviewer’s comments:

Reviewer 2

Thanks for this opportunity to review this manuscript. I appreciate the author's contribution to the field and believe that the paper is valuable and provides new insights into the subject matter.

This manuscript mainly talking about three zone of Gura Rosiei tailing dams and itself mechanical property and hydrogeology element which decide the influence of environment.

The tailing dam are the Special structures, and the mechanical property are complicated. This manuscript utilizing the Fellenius, Taylor, Bishop, Morgenstern-Price, Spencer, Janbu, and Finite Element Method to solve and analyze the stability and water-proof performance. Therefore, to draw the conclusion of stability, risk occurrence probability and eco-friendly redevelopment.

The manuscript is well-structured, well-written, and the experimental design is sound. However, there are a few areas that could be improved:

  1.  The results and the effort by authors didn’t embody in the abstract. The reader couldn’t catch the key content fast and accurately

Response: Esteemed reviewer, Thank you very much for your constructive suggestion. According to the review comments, the abstract is as follows:

The tailing dams are special constructions that are part of a complex of works related to the installations for preparing mining masses. These constructions have the role of mechanical treatment of waste water and the safe storage of sterile resulting from ore processing. The closing and greening of tailings dams is done taking into account the general stability of the pond and its related constructions, as well as the integration into the surrounding environment of the surfaces occupied by mining waste deposits (tailing dams). This study presents the results of the hydrogeotechnical and stability study carried out on the Gura RoÅŸiei tailing dam location. This analysis aims to evaluate the stability degree of the three compartments that make up the tailings deposit in order to carry out the closing and greening works of the tailing dam, and to conclude whether the idea of raising them by 1.5 ... 2 m with tailings is feasible. This study was based on field observations, geotechnical drilling, and physical-chemical analyses of the collected samples. Due to the shallow depth, the drilling could not highlight a hydrostatic level except for the areas with excess humidity, areas represented by certain lenticular intercalations of sandy dust that yield water very slowly. These areas are not continuous and cannot define a reference hydrostatic level. All situations were analyzed by four different methods that satisfy the static equilibrium of forces or moments (Bishop and Janbu) or simultaneously of forces and moments (Spencer and Morgenstern-Price). From the point of view of the stability calculations performed in the hypothesis in which the three ponds become active for the storage of tailings, assuming a corresponding piezometric level, the resulting safety factors are relatively close to the standard values (Fs ≥ 1.4) for the static analysis, and in seismic conditions they are at the limit of equilibrium. The NE slope of pond no. 2 (section 3-3) shows values below the standard safety limit for this type of work. Also, tailing dam no. 3 resulted from the calculations as totally inadequate for the elevation. From the obtained results, it was found that the location formed by the three compartments that make up the Gura Rosiei tailing dam presents major disadvantages for a future storage of the flotation tailings, being at the same time an imminent danger for the environment. Due to its reduced capacity to release water from the pores, the settled material is still in a saturated state, and it is assumed that the foundation land, made up of the terrace deposits of the Abrud River, is clogged at the interface with the settled material and unable to naturally drain the excess moisture from the dam's body.

  1. The Introduction in chapter 1 didn’t show the vertical view and made readers hard to get the environment, height, the direction of river of the tailing dam. And can’t to know the influence of pollution made by heavy metal etc.;

Response: Thank you very much for your valuable comments. The main source of environmental pollution is, however, acidic water resulting from the exposure of sulphur-containing rocks to the existing atmospheric conditions. This process leads to the formation of sulfuric acid, which dissolves the heavy metals in the rock. Heavy metals are easily transported into groundwater and surface water in the area, affecting aquatic biota and contaminating sediments along watercourses. Contamination of the soil with heavy metals is equally possible but varies according to the distance from the mining works. A very low pH was recorded inside the tailings ponds. On the tailings pond, tailings samples show high lead values, most of them above the alert threshold. These high concentrations in areas exposed to previous mining activities are correlated with the existing geological substrate. The highest concentrations of metals generally coincide with locations where a very acidic pH prevails. In general, it can be confirmed that soil pollution is restricted in the vicinity of the mining area, with very few locations registering values above the alert threshold.

It can be seen from figure 1 the direction of river of the tailing dam.

Figure 1. Situation plan of the tailings dam.

In addition, we provide you some photos that show the condition of the settling pond at the time when we started this study.

Provisional clear water discharge channel

  1. The Introduction in chapter 1 didn’t show the vertical view and made readers hard to get the environment, height, the direction of river of the tailing dam. And can’t to know the influence of pollution made by heavy metal etc.;

Response: Thank you very much for your valuable comments. The maximum height of the deposit in these compartments is approx. 45 m. It was made by the "inward raising" method, and the raising dikes were built from tailings deposited and sedimented in the previous stage. Pond No. 1 has a height of about 15 m, pond No. 2 is about 45 m, and pond No. 3 is about 42 m, together having stored a quantity of almost 10 million tonnes of preparation waste.

  1. The conclusion manuscript given hardly to share in another project and which tailing dam can fit this conclusion didn’t illustrate.

Therefore, this is a valuable manuscript that requires further improvement. I hope the authors will carefully consider my suggestions and pay more attention to the rigor of methods, depth of argumentation, and reliability of results in future studies.

Thank you again to the editor for giving me the opportunity to review this paper. I look forward to seeing the authors achieve even more outstanding results in their future research.

Congratulations on a job well done.

Response: Esteemed reviewer, Thank you very much for your constructive suggestion. Relevant revisions of conclusions are as follows:

From the point of view of the stability calculations performed in the hypothesis in which the three ponds become active for the storage of tailings, assuming a corresponding piezometric level, the resulting safety factors are relatively close to the standard values (Fs 1.4) for the static analysis, and in seismic conditions they are at the limit of equilibrium. In some areas, the slope shows values below the standard safety limit for this type of work. The calculations showed that one of the compartments of the sedimentation pond is totally unsuitable for elevation. From the results obtained, it was found that the location formed by the three compartments that make up the Gura Rosiei settling pond presents major disadvantages for the future storage of the flotation tailings and is at the same time an imminent danger for the environment. A solution for greening the dam in the dry version without the presence of water accumulations on its surface can be provided.

Sincerely yours,

Mihaela Toderas on behalf of all the authors

Reviewer 3 Report

Sustainability-2364378

The authors have presented a stability analysis of the tailings dam for the closing, greening, and ensuring its safety via a case study. The obtained results can be used in the problem of slope stability, especially since it is based on a case study. The following items should be corrected in the manuscript.

1-      In the introduction of the manuscript, several references are presented cumulatively (e.g., example, lines 102 and 116). It is necessary to present the related articles in individual form and to provide a few lines of explanation for each one.

2-      One of the keywords is ‘exfiltration’, which is not in the title and abstract of the article! Additionally, the words ‘mining’ and ‘waste’ should be replaced with the phrase ‘mining waste deposits’.

3-      The manuscript has not yet been accepted and published; it can be called an article, so using the word article in the text is incorrect and can be replaced by ‘study’ (e.g., see line 15).

4-      In the last paragraph of the introduction, it is necessary to present the innovation of the study compared to the previous publications, especially the presented paper in the introduction of the manuscript.

5-      Subsection 3.2 should be moved to a separate section after the introduction under the title of methodology.

6-      The references of Tables 5 and 6 are not provided in the manuscript.

7-      Considering that the results of the slope stability analysis are presented in Table 8 and Figure 13, it is not necessary to present all the figures related to the analysis in Figures 14 to 31. One or two figures examples are sufficient.

8-      The geotechnical information of tailings dam sites should be presented in tabular form in the manuscript.

9-      In section 5.2, information is provided that shows that the slope is also analyzed by numerical method. But this information is not complete and it is necessary to provide the figure of modeling, boundary conditions, variation of suction in the depth, and used soil water retention curve. In this regard see and add: Stochastic analysis of rainfall-induced slope instability and steady-state seepage flow using the random finite-element method.

10-   In professional articles, the most important achievements of the study are presented in the conclusion section. This is while the conclusion part of the manuscript is very long. It is necessary to summarize it very much and reduce it to one-third of the existing amount.

11-   The manuscript is related to the analysis of the stability of the slope in static and seismic modes by limit equilibrium. The following articles are also related to this, considering the effect of soil heterogeneity. They can be added to the introduction of the manuscript. A seismic slope stability probabilistic model based on Bishop's method using analytical approach (2015), Analytical stochastic analysis of seismic stability of infinite slope (2015), System reliability analysis of slopes based on the method of slices using sequential compounding method (2019).

12-   Instead of the word dynamic, use seismic, which includes pseudo-static. Dynamic analysis refers to the analysis using the acceleration-time graph.

Sustainability-2364378

The authors have presented a stability analysis of the tailings dam for the closing, greening, and ensuring its safety via a case study. The obtained results can be used in the problem of slope stability, especially since it is based on a case study. The following items should be corrected in the manuscript.

1-      In the introduction of the manuscript, several references are presented cumulatively (e.g., example, lines 102 and 116). It is necessary to present the related articles in individual form and to provide a few lines of explanation for each one.

2-      One of the keywords is ‘exfiltration’, which is not in the title and abstract of the article! Additionally, the words ‘mining’ and ‘waste’ should be replaced with the phrase ‘mining waste deposits’.

3-      The manuscript has not yet been accepted and published; it can be called an article, so using the word article in the text is incorrect and can be replaced by ‘study’ (e.g., see line 15).

4-      In the last paragraph of the introduction, it is necessary to present the innovation of the study compared to the previous publications, especially the presented paper in the introduction of the manuscript.

5-      Subsection 3.2 should be moved to a separate section after the introduction under the title of methodology.

6-      The references of Tables 5 and 6 are not provided in the manuscript.

7-      Considering that the results of the slope stability analysis are presented in Table 8 and Figure 13, it is not necessary to present all the figures related to the analysis in Figures 14 to 31. One or two figures examples are sufficient.

8-      The geotechnical information of tailings dam sites should be presented in tabular form in the manuscript.

9-      In section 5.2, information is provided that shows that the slope is also analyzed by numerical method. But this information is not complete and it is necessary to provide the figure of modeling, boundary conditions, variation of suction in the depth, and used soil water retention curve. In this regard see and add: Stochastic analysis of rainfall-induced slope instability and steady-state seepage flow using the random finite-element method.

10-   In professional articles, the most important achievements of the study are presented in the conclusion section. This is while the conclusion part of the manuscript is very long. It is necessary to summarize it very much and reduce it to one-third of the existing amount.

11-   The manuscript is related to the analysis of the stability of the slope in static and seismic modes by limit equilibrium. The following articles are also related to this, considering the effect of soil heterogeneity. They can be added to the introduction of the manuscript. A seismic slope stability probabilistic model based on Bishop's method using analytical approach (2015), Analytical stochastic analysis of seismic stability of infinite slope (2015), System reliability analysis of slopes based on the method of slices using sequential compounding method (2019).

12-   Instead of the word dynamic, use seismic, which includes pseudo-static. Dynamic analysis refers to the analysis using the acceleration-time graph.

Author Response

List of Responses

Dear Editors and Reviewers:

Thank you for your letter and for the reviewers’ comments concerning the manuscript entitled “Stability analysis of the tailings dam for the purpose of closing, greening, and ensure its safety. Study case”. Those comments are all valuable and very helpful for revising and improving the paper, as well as providing important guidance for my research. We have studied the comments carefully and have made corrections, which we hope will meet your expectation. We would like to appreciate your insightful and constructive comments on our manuscript. We have fully revised and improved the contents of our paper according to your suggestions. The revised portions are marked in red in the paper. The main corrections in the paper and the responses to the reviewer’s comments are as following:

Responds to the reviewer’s comments:

Reviewer 3

The authors have presented a stability analysis of the tailings dam for the closing, greening, and ensuring its safety via a case study. The obtained results can be used in the problem of slope stability, especially since it is based on a case study. The following items should be corrected in the manuscript.

1 In the introduction of the manuscript, several references are presented cumulatively (e.g., example, lines 102 and 116). It is necessary to present the related articles in individual form and to provide a few lines of explanation for each one.

Response: Esteemed reviewer, Thank you very much for your constructive suggestion. Relevant revisions are as follows. In his studies, Burd [24] analyses changes in tailings thickness and copper levels before, during, and after mining and highlights three distinct impact zones below the dumping depth. The issue of irrational use of natural resources in Ukraine that affects public health, population working ability and macroeconomic performance was study by Koval et al. [25]. The key in the study is the formation of a holistic view of the relationship between pollution and the state of the environment and harm to public health based on the analysis of rational nature management and environmental pollution and their negative impact on environmental health [25]. In [27] contribution possibilities and limitations of such models are discussed with the purpose to give geotechnical engineers (rather than researchers) guidelines to properly select soil models and their corresponding parameters to be used in the finite element method for engineering applications. The frequency of accidents due to the instability of the waste dump slope structures has also increased, resulting in significant fatalities, apart from the economic, social and environmental impacts of these disasters. Numerous scientific studies have been conducted to reduce the occurrence of such incidents [31]. Please see also on the page 9 in the manuscript.

2 One of the keywords is ‘exfiltration’, which is not in the title and abstract of the article! Additionally, the words ‘mining’ and ‘waste’ should be replaced with the phrase ‘mining waste deposits’.

Response: Esteemed reviewer, Thank you very much for your constructive suggestion. The words ‘mining’ and ‘waste’ was replaced with the phrase ‘mining waste deposits’:

3 The manuscript has not yet been accepted and published; it can be called an article, so using the word article in the text is incorrect and can be replaced by ‘study’ (e.g., see line 15).

Response: Esteemed reviewer, Thank you very much for your constructive suggestion. The word “article” in the text was replaced by “study”.

4 In the last paragraph of the introduction, it is necessary to present the innovation of the study compared to the previous publications, especially the presented paper in the introduction of the manuscript.

Response: Esteemed reviewer, Thank you very much for your constructive suggestion. This analysis aims to evaluate the stability degree of the three compartments that make up the tailings deposit in order to carry out the closing and greening works of the tailing dam, and to conclude whether the idea of raising them by 1.5 ... 2 m with tailings is feasible. From the point of view of the stability calculations performed in the hypothesis in which the three ponds become active for the storage of tailings, assuming a corresponding piezometric level, the resulting safety factors are relatively close to the standard values (Fs 1.4) for the static analysis, and in seismic conditions they are at the limit of equilibrium. The NE slope of pond no. 2 (section 3-3) shows values below the standard safety limit for this type of work. Also, tailing dam no. 3 resulted from the calculations as totally inadequate for the elevation. From the obtained results, it was found that the location formed by the three compartments that make up the Gura Rosiei tailing dam presents major disadvantages for a future storage of the flotation tailings, being at the same time an imminent danger for the environment.

5 Subsection 3.2 should be moved to a separate section after the introduction under the title of methodology.

Response: Esteemed reviewer, Thank you very much for your constructive suggestion. Subsection 3.2 was moved. Please see page 8 in the manuscript

6 The references of Tables 5 and 6 are not provided in the manuscript.

Response: Esteemed reviewer, Thank you very much for your constructive suggestion. The references for table 5 was provided.

The matrix in table 6 is created by the authors, based on the information presented in tables 4 and 5 and according to the actual conditions found at the Gura Rosiei tailing dam. The assessment of potential risks associated with major accidents was performed in accordance with the methodology based on consequences, which assesses the consequences of risks, without explicit quantification of occurrence probability of these risks. Table 6 presents the matrix of potential risks. In the matrix of potential risks, the two considered variables, namely probability and severity are classified qualitatively.

Existing measures or those that will need to be implemented in order to have an appropriate level of safety are determined according to frequencies and consequences. In order to assess the potential mining risks associated with the conditions of Gura Rosiei tailing dam, numerical values were assigned for each probability level of potential risk manifestation and for each level of severity. The thick black line in the potential risk matrix is the extent to which the probability of these identified risks should be maintained; in these situations, procedures and work processes must be followed, which means that no additional risk reduction measures would be needed. In the yellow zone, the risks could be reduced to the lowest level considered tolerable, but this reduction involves identifying and implementing of necessary technical or organizational measures. For all risks with frequencies in the red zone, immediate implementation of technical measures is required, using all available resources to reduce the level of risk (s).

7 Considering that the results of the slope stability analysis are presented in Table 8 and Figure 13, it is not necessary to present all the figures related to the analysis in Figures 14 to 31. One or two figures examples are sufficient.

Response: Esteemed reviewer, Thank you very much for your constructive suggestion. Figures 14–31 have been replaced with figures 14–18.

8 The references of Tables 5 and 6 are not provided in the manuscript.

Response: Esteemed reviewer, Thank you very much for your constructive suggestion. The input geotechnical parameters of the materials in the tailings dam body used in the calculations are shown in table 7..

9 In section 5.2, information is provided that shows that the slope is also analyzed by numerical method. But this information is not complete and it is necessary to provide the figure of modeling, boundary conditions, variation of suction in the depth, and used soil water retention curve. In this regard see and add: Stochastic analysis of rainfall-induced slope instability and steady-state seepage flow using the random finite-element method.

Response: Esteemed reviewer, Thank you very much for your constructive suggestion. Since the modelling was done in the GeoStudio SEEP/W demo, which was only available for a period of 30 days, we only took over the results. The analysis was done in January, and due to the expiration of the usage period, we do not have the other information and do not have the funds currently needed to purchase the software.

10 In professional articles, the most important achievements of the study are presented in the conclusion section. This is while the conclusion part of the manuscript is very long. It is necessary to summarize it very much and reduce it to one-third of the existing amount.

Response: Esteemed reviewer, Thank you very much for your constructive suggestion. Relevant revisions are as follows: The location formed by the three compartments that make up the Gura Rosiei tailing dam presents major disadvantages for a future storage of flotation tailings, being at the same time an imminent danger for the environment. From the point of view of the stability calculations performed in the hypothesis in which the three ponds become active for the storage of tailings, assuming a corresponding piezometric level, the resulting safety factors are relatively close to the standard values (Fs 1.4) for the static analysis, and in seismic conditions they are at the limit of equilibrium. In some areas, the slope shows values below the standard safety limit for this type of work. The calculations showed that one of the compartments of the sedimentation pond is totally unsuitable for elevation. From the results obtained, it was found that the location formed by the three compartments that make up the Gura Rosiei settling pond presents major disadvantages for the future storage of the flotation tailings and is at the same time an imminent danger for the environment. A solution for greening the dam in the dry version without the presence of water accumulations on its surface can be provided.

11 The manuscript is related to the analysis of the stability of the slope in static and seismic modes by limit equilibrium. The following articles are also related to this, considering the effect of soil heterogeneity. They can be added to the introduction of the manuscript. A seismic slope stability probabilistic model based on Bishop's method using analytical approach (2015), Analytical stochastic analysis of seismic stability of infinite slope (2015), System reliability analysis of slopes based on the method of slices using sequential compounding method (2019).

Response: Esteemed reviewer, Thank you very much for your constructive suggestion. Relevant revisions are as follows: Johari et al. (2015) proposed two probabilistic models to assess the effect of seismic loading. They develop a probabilistic model of seismic slope stability based on Bishop's method using JDRV method and make a comparison of the probability density func-tions of slope safety factor with the Monte Carlo simulation [54]. The same Monte Carlo Simulation was used by the author [55] to compare the results obtained applying the jointly distributed random variables method for stochastic analysis and reliability assessment of seismic stability of infinite slopes without seepage. Considering the method of slices and by selecting stochastic soil parameters, the most critical failure surface is determined by the particle swarm optimization algorithm. The results are verified by Monte Carlo Simulation (MCS); however, this method is quite time-consuming for this purpose [56].

12 Instead of the word dynamic, use seismic, which includes pseudo-static. Dynamic analysis refers to the analysis using the acceleration-time graph.

Response: Esteemed reviewer, Thank you very much for your constructive suggestion. Instead of the word "dynamic", the word "seismic" was used in the whole manuscript, including the figures.

Sincerely yours,

Mihaela Toderas on behalf of all the authors

Round 2

Reviewer 3 Report

The authors applied all comments in an appropriate manner. The revised manuscript is acceptable for publication.

The authors applied all comments in an appropriate manner. The revised manuscript is acceptable for publication.